# Dynamic Semantic-Aware Correlation Modeling for UAV Tracking

**Xinyu Zhou**[1*]   **Tongxin Pan**[1*]   **Lingyi Hong**[1]   **Pinxue Guo**[2]   **Haijing Guo**[1]

**Zhaoyu Chen**[2]   **Kaixun Jiang**[2]   **Wenqiang Zhang**[1,2†]
[1]College of Computer Science and Artificial Intelligence, Fudan University
[2]College of Intelligent Robotics and Advanced Manufacturing, Fudan University
{zhouxinyu20,wqzhang}@fudan.edu.cn, kamipantx@gmail.com

## Abstract

UAV tracking can be widely applied in scenarios such as disaster rescue, environmental monitoring, and logistics transportation. However, existing UAV tracking methods predominantly emphasize speed and lack exploration in semantic awareness, which hinders the search region from extracting accurate localization information from the template. The limitation results in suboptimal performance under typical UAV tracking challenges such as camera motion, fast motion, and low resolution, etc. To address this issue, we propose a dynamic semantic aware correlation modeling tracking framework. The core of our framework is a Dynamic Semantic Relevance Generator, which, in combination with the correlation map from the Transformer, explore semantic relevance. The approach enhances the search region's ability to extract important information from the template, improving accuracy and robustness under the aforementioned challenges. Additionally, to enhance the tracking speed, we design a pruning method for the proposed framework. Therefore, we present multiple model variants that achieve trade-offs between speed and accuracy, enabling flexible deployment according to the available computational resources. Experimental results validate the effectiveness of our method, achieving competitive performance on multiple UAV tracking datasets. The code is available at https://github.com/zxyyxzz/DSATrack.

## 1 Introduction

Visual object tracking[55, 9, 29, 28, 59] is a fundamental and important task in computer vision. In recent years, unmanned aerial vehicles (UAV) technology has rapidly developed and demonstrated broad application in various fields such as military, transportation, environmental monitoring, and disaster relief [43]. However, to achieve autonomous task execution in complex environments, object tracking is a crucial component. UAV tracking must address various challenges in dynamic scenarios.

The early UAV tracking methods [36, 31, 35, 27] employed a correlation filter to enhance the discriminative capability, enabling location and tracking. However, these correlation filter-based methods lacked robust representation capabilities. Consequently, UAV tracking methods based on convolutional neural networks (CNNs) [4, 49, 52] emerged, utilizing Siamese network to enhance feature representation, significantly improving tracking accuracy. With the rise of Transformers [55, 11, 6, 51], UAV tracking based on one-stream framework [37, 34, 32, 21] have been studied. Most of the methods focus on improving the efficiency of UAV tracking, while neglecting exploration

---

[*]Equal contribution.
[†]Corresponding author.

39th Conference on Neural Information Processing Systems (NeurIPS 2025).

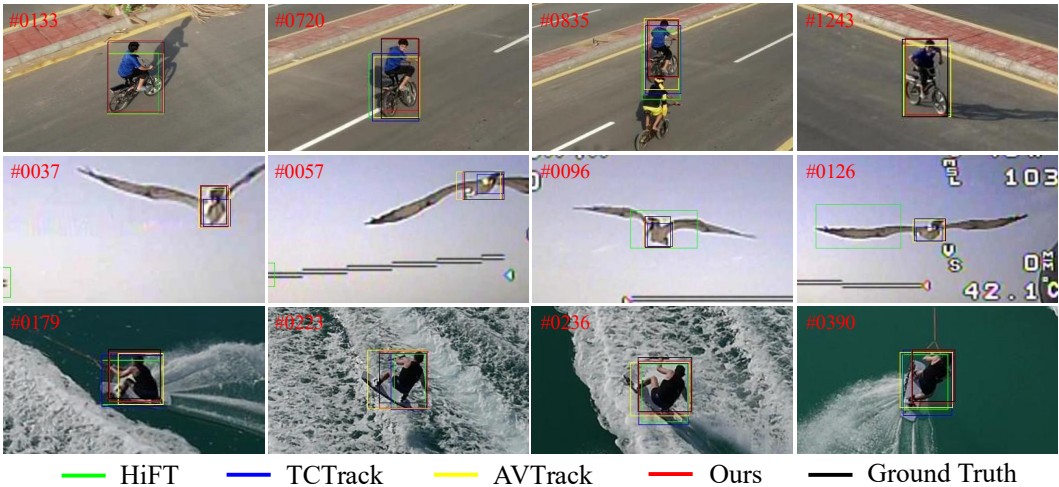

| —— HiFT | —— TCTrack | —— AVTrack | —— Ours | —— Ground Truth |

Figure 1: Comparison with SOTA methods under typical UAV challenges. The first to third rows represent camera motion, fast motion, and low resolution, respectively.

of the essence of tracking. Correlation filter-based methods and CNN-based methods rely on local modeling, while Transformer-based methods utilize global modeling. All of the approaches overlook the relevance between semantic features, as shown in Figure 1, leading to suboptimal performance in UAV scenarios, such as camera motion, fast motion, and low resolution, etc.

To address this issue, we propose a dynamic semantic-aware correlation modeling method and develop a UAV tracking framework based on the approach, named DSATrack. Specifically, we use the search region and template&patches as inputs and employ Transformer blocks to achieve semantic matching, semantic awareness, and feature extraction and fusion. Importantly, we design a Dynamic Semantic Relevance Generator within the Transformer to dynamically achieve Semantic Aware Modeling . By leveraging the Semantic Aware Modeling to fuse similar semantic features, our method alleviates semantic ambiguities. The semantic-aware correlation map further facilitates the search region in extracting representative features from the template&patches, thereby enhancing the perception and localization capability of UAV tracking and mitigating tracking drift. Finally, to improve the tracking efficiency, we design a pruning technique tailored to the characteristics of DSATrack. By selectively pruning Transformer blocks, we retain the tracking precision to the greatest extent while achieving high speed tracking. We provides multiple variants for different UAV Tracking application scenarios. For instance, offline tasks such as aerial video prioritize tracking accuracy, while real-time applications like autonomous target following demand fast response times.

Our contributions can be summarized as follows.

• We propose a dynamic semantic-aware UAV tracking framework, DSATrack, which utilizes a Dynamic Semantic Relevance Generator to produce semantic-aware correlation map. These maps are used to fuse features with similar semantic features, effectively reducing tracking drift.

•To further optimize DSATrack, we design a pruning technique that selectively removes Transformer blocks with minimal impact on accuracy. This approach achieves a balance between precision and speed, enabling flexible deployment according to the available computational resources.

•We conduct extensive experiments, demonstrating that DSATrack achieves state-of-the-art performance. Additionally, ablation studies validate the effectiveness of the proposed methods.

## 2 Related Works

### 2.1 UAV Tracking

Correlation filter-based methods, such as KCF [25] and DSST [13], were among the earliest applied to UAV tracking due to their high efficiency, achieving strong performance in speed and accuracy.

However, they struggle under camera motion or fast motion, which are common in UAV scenarios. With the rise of deep learning, Siamese network-based trackers—such as TransT [9], AiATrack [19], STARK [53], ToMP [41], and KeepTrack [42]—have significantly improved tracking robustness. TCTrack [4] further adapts Siamese models to UAV tracking. Yet, their heavy computational demands hinder real-time UAV deployment. Recently, one-stream Transformer frameworks [55, 11, 8, 6, 1, 54, 20] have emerged, offering enhanced modeling of long-range dependencies with improved efficiency. Trackers like TATrack [33] and AVTrack [37] utilize this framework to better capture interactions between template and search regions under UAV-specific challenges. Although LoReTrack [14] is not tailored for UAV tracking and lacks real-world UAV evaluation, it demonstrates promising results on a UAV dataset using a lightweight one-stream design. In contrast, we propose a Dynamic Semantic-Aware Correlation Modeling approach within a one-stream Transformer framework to address key UAV challenges such as camera motion, fast motion, and low resolution.

## 2.2 Semantic Correspondence

In visual perception tasks, semantic correspondence is typically categorized into two types: explicit correspondence and implicit correspondence. Explicit correspondence involves semantic-aware modeling through supervised loss functions, while implicit correspondence does not rely on direct supervision for semantic correspondence. Explicit correspondence methods, such as contrastive learning [7, 24], Triplet Loss [46], and InfoNCE [45], supervise the semantic relationships between images or pixels by pulling closer those belonging to the same category and pushing apart those of different categories. This enables models to learn semantic relationships, improving the accuracy of downstream tasks [22, 58]. On the other hand, implicit correspondence is commonly employed in tasks like classification [40], detection [61], segmentation [10] and Tracking [19, 9] using Transformer-based architectures [48, 15]. These architectures implicitly learn semantic relationships among image features through task-specific losses. Specifically, Transformers compute a correlation map using queries and keys [5, 47], where the correlation map represents the semantic similarity between features [12]. However, in implicit semantic correspondence methods, there is limited exploration into explicitly optimizing semantic correspondence, which often results in suboptimal modeling of the correlation map. This lack of optimization can hinder the model's ability to capture semantic relationships. Therefore, we propose a dynamic semantic-aware correlation modeling method for UAV tracking to enhance semantic perception capabilities, improving tracking accuracy.

## 3 Method

In this section, we will first present our proposed tracking framework, DSATrack. Next, we will describe the Dynamic Semantic-aware Transformer. Following that, we will detail the pruning method designed specifically for DSATrack. Finally, we will present description of the prediction head.

### 3.1 Overall Framework

The overall framework of the proposed DSATrack is shown in Figure 2. The entire tracking framework consists of three components: input representation, feature matching and interaction, and prediction.

**Input Representation**. The input of DSATrack is the Template & Patches $z \in \mathbb{R}^{n \times 3 \times H_z \times W_z}$ and Search Region $x \in \mathbb{R}^{3 \times H_x \times W_x}$ , where $H$ and $W$ represent the height and width of the Template and Search Region. The Template & Patches consist of a fixed template$\in \mathbb{R}^{1 \times H_z \times W_z}$ and several patches$\in \mathbb{R}^{(n-1) \times H_z \times W_z}$ that exhibit representative object characteristics. The Patches are generated and updated over time steps in exactly the same manner as RFGM [60]. Subsequently, the inputs are projected into tokens through a Patch Embedding. **Feature matching and Interaction**. The generated Template Tokens $T_z \in \mathbb{R}^{n \times 3 \times h_z \times w_z}$ and Search Tokens $T_x \in \mathbb{R}^{n \times 3 \times h_x \times w_x}$ are then fed into the Transformer Blocks for feature matching and interaction, where $h$ and $w$ represent the height and width of the features after a $16\times$ downsampling. The specific computation method of the Transformer block is consistent with OSTrack [55]. Importantly, we propose a Dynamic Semantic-aware Transformer for semantic matching and feature fusion. Specifically, we designed a Dynamic Semantic Relevance Generator to construct Semantic Relevance. Using the Semantic Aware Modeling, we fuse the similarities of features with similar semantics to obtain an optimized Semantic-Aware Correlation Map. Based on this correlation map, unimportant template tokens are discarded. The filtered Template Tokens, Search Tokens, and the Semantic-Aware Correlation Map

are then fed into a Hybrid Attention module to facilitate feature interaction. **Prediction**. Finally, the Search Tokens are reassembled into an image and passed to the prediction head, which predicts the bounding box of the object.

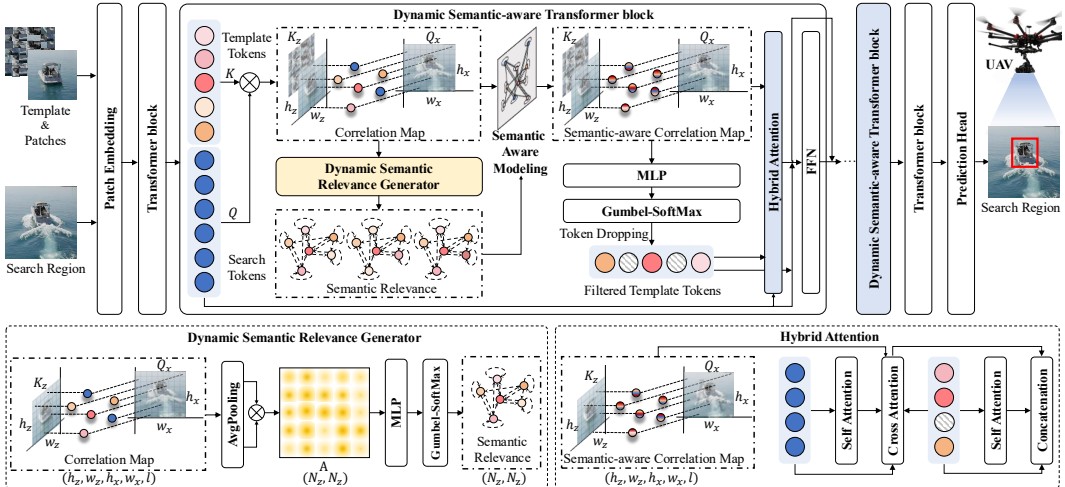

Figure 2: **The framework of DSATrack**. The figure above illustrates the overall pipeline of DSATrack, while the two figures below provide a detailed explanation of the Dynamic Semantic Relevance Generator and the Hybrid Attention, respectively.

## 3.2 Dynamic Semantic-aware Transformer

The Template Tokens $T_z \in \mathbb{R}^{n \times h_z \times w_z \times d_k}$ and Search Tokens $T_x \in \mathbb{R}^{1 \times h_x \times w_x \times d_k}$ are inputs of Dynamic Semantic-aware Transformer, where $d_k$ is the dimensionality of the features. For simplicity and clarity, we present the details using single template, $T_z \in \mathbb{R}^{1 \times h_z \times w_z \times d_k}$. We use a linear layer to transform the Search Tokens and Template Tokens into ($Q_x \in \mathbb{R}^{h_x \times w_x \times d_k}$) and ($K_z \in \mathbb{R}^{h_z \times w_z \times d_k}$). As shown in Figure 2, we compute the Correlation Map $C$ between the $Q_x$ and $K_z$ as:

$$C_{zx} = \frac{Q_x K_z^\top}{\sqrt{d_k}} \in \mathbb{R}^{h_x \times w_x \times h_z \times w_z \times l}, \tag{1}$$

where $C_{zx}$ captures the correlation scores between query from $x$ and key from $z$, corresponding to the $z$-to-$x$ similarity. $l$ represents the number of heads in the Transformer.

In the Correlation Map $C_{zx}$, the modeling of the similarity relationship between points on $Q_x$ and points on $K_z$ can easily lead to ambiguity. We employ a dynamic perception-based prediction approach to calculate and fuse the similarity between semantically related points on $K_z$ and $Q_x$.

**Dynamic Semantic Relevance Generator**. We first reshape $C_{zx} \in \mathbb{R}^{h_x \times w_x \times h_z \times w_z \times l}$ into $C_{zx} \in \mathbb{R}^{N_x \times N_z \times l}$, where $N_x = h_x \times w_x$ and $N_z = h_z \times w_z$. Then, We propose a Semantic Aware Modeling $\mathcal{G} = (\mathcal{V}, \mathcal{E})$, where $\mathcal{V} = \{\vec{c}_i \mid \vec{c}_i \in C_{zx}, 1 \leq i \leq N_z\}$ represents the set of nodes and the shape of $\vec{c}_i$ is $N_x \times 1 \times l$. In other words, we retrieve nodes along the second dimension of $C_{zx}$. $\mathcal{E} = \{(u, v) \mid \forall u \in \mathcal{V}, \forall v \in \mathcal{V}\}$ represents the set of Semantic Relevance. Specifically, we apply average pooling to the node features and we concatenate the vector $\vec{c}_i'$ to form $\vec{v}$:

$$\vec{c}_i{}' = \frac{\sum_{j=0}^{N_x} \vec{c}_j}{N_x}, \vec{c}_j \in \mathbb{R}^{1 \times 1 \times l} \subseteq \vec{c}_i, \tag{2}$$

$$\vec{v} = [\vec{c_1}', \vec{c_2}', ..., \vec{c_{N_z}}'] \in \mathbb{R}^{N_z \times 1 \times l}, \tag{3}$$

where $[\cdot]$ represents concatenation.

Subsequently, we compute the similarity score $A$ between nodes via matrix multiplication:

$$A = (\vec{v})^\top (\vec{v}), A \in \mathbb{R}^{N_z \times N_z \times l}. \tag{4}$$

After that, a tiny prediction network consisting of a multi-layer perceptron(MLP) and Log-Softmax function is utilized to predict the possibility that each Relevance will be retained and discarded:

$$\pi_{ij} = \log \frac{\exp\left(\text{MLP}\left(A_{ij}\right)\right)}{\sum_{k=1}^{N_z} \exp\left(\text{MLP}\left(A_{ik}\right)\right)} \in \mathbb{R}^2, 1 \leq i, j \leq N_z. \tag{5}$$

where $\pi_{ij,0}$ is the probability of retaining the Relevance $(i,j)$ and $\pi_{ij,1}$ is probability of removing it.

Next, we apply the Gumbel-Softmax to produce the Dynamic Semantic Relevance $\mathcal{E}$, making the $\pi$ differentiable and retaining the most relevant connections:

$$\mathcal{E} = \text{Gumbel} - \text{Softmax}\left(\pi\right) \in \{0,1\} \in \mathbb{R}^{N_z \times N_z}. \tag{6}$$

**Semantic Aware Modeling**. Therefore, we utilize the Semantic Aware Modeling to optimize the Correlation Map, as defined by the following formula:

$$C'_{zx} = \Lambda^{-\frac{1}{2}} \hat{\mathcal{E}} \Lambda^{-\frac{1}{2}} C_{zx}^{\top} W_v \in \mathbb{R}^{N_x \times N_z \times l}, \tag{7}$$

where $\hat{\mathcal{E}} = \mathcal{E} + I$, $I$ is the identity matrix, and $\Lambda_{ii} = \sum_j \mathcal{E}_{ij}$ is the degree matrix.

**Hybrid Attention**. The Semantic-aware Correlation Map is fed into an MLP and Gumbel-softmax to evaluate the importance of the Template Tokens. Less significant tokens are discarded based on $\mathcal{M} \in \{0,1\} \in \mathbb{R}^{N_z \times N_z}$ to accelerate computation:

$$\mathcal{M} = \text{Gumbel} - \text{Softmax}\left(\text{Log} - \text{Softmax}(\text{MLP}(C'_{zx}))\right). \tag{8}$$

Subsequently, we combine $Q_z$, $Q_x$, $K_z$, $K_x$, $V_z$, $V_x$, and $C'_{zx}$ to compute the Hybrid Attention:

$$\begin{aligned} &\text{HAttention}(Q_x, K_x, V_x, K_z, V_z) = \\ &[\text{SelfAttention}(Q_z, K_z, V_z), \text{SelfAttention}(Q_x, K_x, V_x), \\ &\text{CrossfAttention}(Q_x, K_z, V_z)]. \end{aligned} \tag{9}$$

The formula for Self-attention is as follows:

$$\text{SelfAttention}(Q, K, V) = \text{SoftMax}(\frac{QK^{\top}}{\sqrt{d_k}})V. \tag{10}$$

Cross-Attention can be calculated as follows:

$$\text{CrossAttention}(Q_x, K_z, V_z) = \text{SoftMax}(C'_{zx} + C_{zx})V_z. \tag{11}$$

Finally, we incorporated a Feedforward Neural Network (FFN) to enhance the model's fitting ability. The Dynamic Semantic-aware transformer can be summarized as follows:

$$T''_{xz} = T'_{xz} + \text{FFN}(T'_{xz}), \tag{12}$$

$$T'_{xz} = \text{HAttention}(Q_x, K_x, V_x, K_z, V_z) + T_{xz}, \tag{13}$$

where $T_{xz}$ represents the concatenation of $T_z$ and $T_x$.

### 3.3 Hierarchical Contribution Ranking Pruning

We introduce the hierarchical contribution metric $\delta_i$, which quantifies the role of each Transformer block in the feature transformation process of DSATrack. The definition of hierarchical contribution is based on the similarity between the hidden state of each layer and its successor. If the input of a layer has a high similarity with the input of the next layer, it indicates that the feature transformation of this layer has a minimal effect on the subsequent layer's input, resulting in a lower contribution. Conversely, if the similarity is low, it suggests that this layer plays a significant role.

The calculation formula for the hierarchical contribution $\delta_i$ of the $i$-th Transformer layer is as follows:

$$\delta_i = 1 - \frac{1}{M} \sum_{m=1}^{M} \cos(T_{i,m}, T_{i+1,m}) = 1 - \frac{1}{M} \sum_{m=1}^{M} \frac{T_{i,m} \cdot T_{i+1,m}}{\|T_{i,m}\|_2 \cdot \|T_{i+1,m}\|_2}, \tag{14}$$

where $T_{i,m}$ is the $m$-th hidden state of $i$-th Transformer block, and $M$ is the number of hidden states.

The model consists of 12 Transformer blocks, with two types: Dynamic Semantic-Aware Transformer blocks located at layers $\mathcal{D} = \{4, 7, 10\}$, and Standard Transformer blocks in the remaining layers $\mathcal{S} = \{1, 2, 3, 5, 6, 8, 9, 11, 12\}$. For the Dynamic Semantic-Aware Transformer blocks and the Standard Transformer blocks, we first sort the contribution scores $\delta_i$ separately for each type of block:

$$\mathcal{C}_{\mathcal{D}} = \text{Sort}(\{\delta_i \mid i \in \mathcal{D}\}), \quad \mathcal{C}_{\mathcal{S}} = \text{Sort}(\{\delta_i \mid i \in \mathcal{S}\}). \tag{15}$$

Based on these sorted values, we select the lowest contributing blocks for pruning. The pruning sets for each type of block are determined by pruning ratios $p_d$ and $p_s$ as follows:

$$\begin{aligned}
\mathcal{D}_{\text{prune}} &= \{i \mid i \text{ is the lowest } \lceil p_d \cdot |\mathcal{D}| \rceil \text{ layers in } \mathcal{C}_{\mathcal{D}}\}, \\
\mathcal{S}_{\text{prune}} &= \{i \mid i \text{ is the lowest } \lceil p_s \cdot |\mathcal{S}| \rceil \text{ layers in } \mathcal{C}_{\mathcal{S}}\}.
\end{aligned} \tag{16}$$

These pruned blocks are then discarded, and the remaining blocks from both sets are combined to form the new pruned model. The remaining Dynamic Semantic-Aware Transformer blocks and Standard Transformer blocks are:

$$\mathcal{D}_{\text{new}} = \mathcal{D} \setminus \mathcal{D}_{\text{prune}}, \quad \mathcal{S}_{\text{new}} = \mathcal{S} \setminus \mathcal{S}_{\text{prune}}. \tag{17}$$

Finally, the new model's layer set after pruning is:

$$\mathcal{L}_{\text{new}} = \mathcal{D}_{\text{new}} \cup \mathcal{S}_{\text{new}}. \tag{18}$$

This process allows for a balance between model efficiency and performance, with the pruning ratio $p_d$ and $p_s$ adjustable to optimize computational speed and storage requirements.

## 3.4 Prediction Head

Following OSTrack [55], we construct our prediction head to estimate the bounding box from the transformer-generated search region features. The head consists of three branches: one predicts the classification score map $S \in [0, 1]^{h_x \times w_x}$, another predicts the offset $O \in [0, 1)^{2 \times h_x \times w_x}$, and the third predicts the normalized box size $E \in [0, 1]^{2 \times h_x \times w_x}$. The location with the highest score, $(x_d, y_d) = \arg\max_{(x,y)} S_{xy}$, is combined with $O$ and $E$ to generate the final bounding box.

$$x = x_d + O(0, x_d, y_d), y = y_d + O(1, x_d, y_d), w = E(0, x_d, y_d), h = E(1, x_d, y_d). \tag{19}$$

Table 1: The hierarchical contribution of DSATrack-ViT-B transformer layers on UAV123 benchmark.

| Dataset | Layer 2 | Layer 3 | Layer 4 | Layer 5 | Layer 6 | Layer 7 | Layer 8 | Layer 9 | Layer 10 | Layer 11 | Layer 12 |
|---------|---------|---------|---------|---------|---------|---------|---------|---------|----------|----------|----------|
| UAV123 | 0.1325 | 0.0739 | 0.0615 | 0.0581 | 0.0429 | 0.0457 | 0.0438 | 0.0358 | 0.0428 | 0.0657 | 0.1444 |

Table 2: Setup of removed layers for the pruned variants.

| Variant | Removed Layers | Variant | Removed Layers |
|---------|----------------|---------|----------------|
| DSATrack-D8 | 6, 7, 9, 10 | DSATrack-D6 | 5, 6, 7, 8, 9, 10 |
| DSATrack-D7 | 6, 7, 8, 9, 10 | DSATrack-D4 | 3, 5, 6, 7, 8, 9, 10, 11 |

## 4 Experiments

### 4.1 Implementation Details

**Model**. DSATrack employs the ViT-B [15] as the backbone, initialized with MAE [23], referred to as DSATrack-ViT-B. For the pruned variants of DSATrack, we consider four configurations with 8, 7, 6, and 4 transformer layers, named DSATrack-D8, DSATrack-D7, DSATrack-D6, and DSATrack-D4. All pruned models apply the pruning ratio $p_d = \frac{2}{3}$. The pruning ratios $p_s$ are set to $\frac{2}{9}$, $\frac{3}{9}$, $\frac{4}{9}$ and $\frac{6}{9}$ for the four variants. Table 1 presents the hierarchical contribution of DSATrack-ViT-B transformer layers on UAV123 , yielding the layer removal setup for pruned model variant, as shown in Table 2. In addition, we provide versions based on ViT-Tiny and DeiT-Tiny, named DSATrack-ViT-T and DSATrack-DeiT-T. Templates are resized to $128 \times 128$, while Search Region are resized to $256 \times 256$.

**Training**. We train the model on TrackingNet [44], GOT-10k [26], LaSOT [17], and COCO [39]. The training process consists of two stages. In the first stage, the model is trained for 300 epochs with 3 templates. The learning rate for the Dynamic Semantic-aware Transformer and prediction head is set to $4 \times 10^{-4}$, the learning rate for the remaining parameters is set to $4 \times 10^{-5}$. At epoch 240, the learning rate is decayed by a factor of 10. In the second stage, the model is finetuned for 50 epochs with 6 templates. AdamW optimizer is used with a weight decay of $10^{-4}$. The loss function comprises the L1 loss, Focal loss, GIoU loss and ratio loss, consistent with RFGM [60]. All training tasks are conducted on 4 NVIDIA GeForce RTX 3090 GPUs with a batch size of 24.

**Inference**. For fairness, the speed tests of other models and our model are conducted on the same NVIDIA RTX 3090 GPU without data loading overhead. The size of our Templates & Patches is set to $3 \times N_z$. The update interval of Patches is set to 5 for $t \le 100$, doubled every 100 frames until t = 500, and then remains 160. The interval doubles every 100 frames until it reaches 160, where it remains fixed beyond 160 frames. At the 4th, 7th, and 10th layers, the number of retained template tokens is set to $\lfloor 3 \times N_z \times 0.9 \rfloor$, $\lfloor 3 \times N_z \times 0.8 \rfloor$, and $\lfloor 3 \times N_z \times 0.7 \rfloor$, $\lfloor \cdot \rfloor$ denotes the floor operation.

## 4.2 State-of-the-Art Comparisons

Figure 5 presents a comparison of DSATrack speed and accuracy against the SOTA methods.

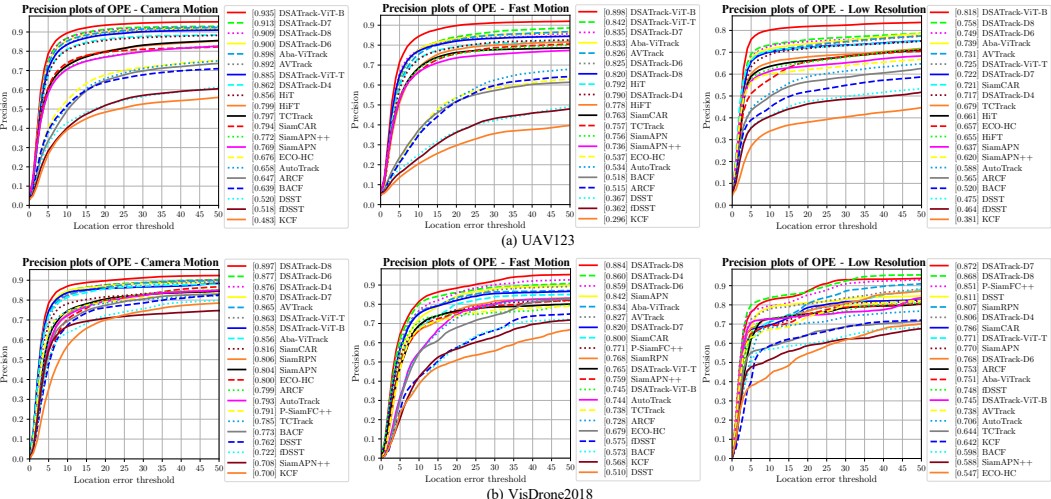

Figure 3: Comparison of state-of-the-art methods under representative UAV challenges.

**DTB70**. DSATrack outperformed SOTA methods. Specifically, DSATrack-ViT-B attained the highest precision (91.2%) and success (70.6%). Notably, even the lightweight variants, DSATrack-D8 and DSATrack-D7, achieved second (89.4% precision) and third place (88.1% precision), demonstrating the effectiveness of our method across different model complexities. DSATrack-VIT-T acquired 86.6% in Precison and 67.4 % in on success, which is outperform the sota method by 2% to 3%. The results highlight the robustness of dynamic semantic-aware correlation modeling in UAV tracking.

**UAVDT**. DSATrack-ViT-B achieved SOTA results with precision and success of 88.1% and 66.3%. DSATrack-D8 also achieved competitive performance, ranking second in precision (85.7%) and achieving the third-best success (63.4%). Importantly, DSATrack-D7 exhibited strong success metrics (62.1%), outperforming prior works such as TATrack and AVTrack. Besides, DSATrack-ViT-T and DSATrack-DeiT-T achieved better performance than SOTA methods, such as AVTrack and LightFC. These results underscore the scalability of DSATrack across varying depths and complexities.

**VisDrone2018**. DSATrack-D8 and DSATrack-ViT-B achieving the top two rankings for success, 69.1% and 67.0%. D7, ViT-T and DeiT-T achieved competitive results, validating the efficacy of our framework in challenging conditions. Compared to existing SOTA methods, DSATrack consistently presented superior semantic-aware modeling, reflecting its adaptability to UAV scenarios. Moreover, the second row of the Figure 3 demonstrates that the pruned version of DSATrack also exhibits significant advantages in addressing the representative challenges of UAV tracking.

Table 3: Comparisons with SOTA methods on UAV tracking datasets.

| Method | Source | DTB70 | | UAVDT | | VisDrone2018 | | UAV123 | |
|---|---|---|---|---|---|---|---|---|---|
| | | Prec. | Succ. | Prec. | Succ. | Prec. | Succ. | Prec. | Succ. |
| SiamAPN[18] | ICRA21 | 78.4 | 58.5 | 71.1 | 51.7 | 81.5 | 58.5 | 76.5 | 57.5 |
| SiamAPN++[3] | IROS21 | 78.9 | 59.4 | 76.9 | 55.6 | 73.5 | 53.2 | 76.8 | 58.2 |
| HiFT [2] | ICCV21 | 80.2 | 59.4 | 65.2 | 47.5 | 71.9 | 52.6 | 78.7 | 58.9 |
| P-SiamFC++[49] | ICME22 | 80.3 | 60.4 | 80.7 | 56.6 | 80.1 | 58.5 | 74.5 | 48.9 |
| TCTrack[4] | CVPR22 | 81.2 | 62.2 | 72.5 | 53.0 | 79.9 | 59.4 | 80.0 | 60.4 |
| UDAT[56] | CVPR22 | 80.6 | 61.8 | 80.1 | 59.2 | 81.6 | 61.9 | 76.1 | 59.0 |
| ABDNet[62] | RAL23 | 76.8 | 59.6 | 75.5 | 55.3 | 75.0 | 57.2 | 79.3 | 60.7 |
| DRCI[57] | ICME23 | 81.4 | 61.8 | 84.0 | 59.0 | 83.4 | 60.0 | - | - |
| HiT[30] | ICCV23 | 75.1 | 59.2 | 62.3 | 47.1 | 74.8 | 58.7 | 82.5 | 63.3 |
| DDCTrack[16] | ICPR24 | 79.0 | 51.1 | - | - | 81.2 | 48.7 | 79.1 | 50.1 |
| SMAT[21] | WACV24 | 81.9 | 63.8 | 80.8 | 58.7 | 82.5 | 63.4 | 81.8 | 64.6 |
| LiteTrack[50] | ICRA24 | 82.5 | 63.9 | 81.6 | 59.3 | 79.7 | 61.4 | 84.2 | 65.9 |
| LightFC-ViT[38] | KBS 24 | 82.8 | 64.0 | 83.4 | 60.6 | 82.7 | 62.8 | 84.2 | 65.5 |
| AVTrack-ViT[37] | ICML24 | 81.3 | 63.3 | 79.9 | 57.7 | 86.4 | 65.9 | 84.0 | 66.2 |
| AVTrack-DeiT[37] | ICML24 | 84.3 | 65.0 | 82.1 | 58.7 | 85.9 | 65.4 | 84.8 | 66.8 |
| DSATrack-ViT-B | Ours | **91.2** | **70.6** | **88.1** | **66.3** | 87.1 | 67.0 | **90.2** | **69.4** |
| DSATrack-D8 | Ours | 89.3 | 69.5 | 85.7 | 63.4 | **90.8** | **69.1** | 87.6 | 67.0 |
| DSATrack-D7 | Ours | 88.1 | 68.4 | 84.6 | 62.1 | 88.6 | 68.0 | 87.3 | 66.6 |
| DSATrack-D6 | Ours | 86.6 | 67.3 | 83.5 | 60.6 | 87.0 | 65.7 | 86.9 | 66.0 |
| DSATrack-D4 | Ours | 84.5 | 65.9 | 82.3 | 58.9 | 84.8 | 64.1 | 84.1 | 63.0 |
| DSATrack-ViT-T | Ours | 86.6 | 67.4 | 84.3 | 61.6 | 84.8 | 64.1 | 85.5 | 65.5 |
| DSATrack-DeiT-T | Ours | 84.5 | 65.7 | 85.0 | 62.0 | 82.0 | 60.8 | 84.4 | 64.5 |

**UAV123**. DSATrack-ViT-B set a new performance of 90.2% (precsion) and of 69.4% (success), surpassing existing SOTA trackers. DSATrack-D8/D7 maintaining competitive scores of 87.6% (precision) and 87.3% (precision). Furthermore, D4, ViT-T and DeiT-T achieved satisfactory accuracy, making it suitable for real-time UAV applications. As shown in the first row of the Figure 3, DSATrack exhibits better performance in addressing challenges such as camera motion, fast motion, and low resolution. This demonstrates the flexibility of DSATrack in balancing efficiency and accuracy.

## 4.3 Ablation Study on Correlation Modeling

Table 4: Ablation study on correlation modeling. The best results are highlighted in bold.

| Variant | Method | DTB70 | | UAVDT | | VisDrone2018 | | UAV123 | |
|---|---|---|---|---|---|---|---|---|---|
| | | Prec. | Succ. | Prec. | Succ. | Prec. | Succ. | Prec. | Succ. |
| DSATrack-ViT-B | Baseline | 87.7 | 67.9 | 85.0 | 63.3 | 86.5 | 65.7 | 87.3 | 67.2 |
| | Baseline+SSAM | 88.0 | 68.3 | 85.5 | 62.6 | **87.8** | 66.6 | 88.8 | 68.5 |
| | Baseline+DSAM | **91.2** | **70.6** | **88.1** | **66.3** | 87.1 | **67.0** | **90.2** | **69.4** |

Table 5: Comparisons of Hierarchical Contribution Ranking Pruning and Sequential Pruning.

| Backbone | Method | DTB70 | | UAVDT | | VisDrone2018 | | UAV123 | | FPS |
|---|---|---|---|---|---|---|---|---|---|---|
| | | Prec. | Succ. | Prec. | Succ. | Prec. | Succ. | Prec. | Succ. | |
| D7 | Sequential Pruning | 86.8 | 67.0 | 82.9 | 60.9 | 86.2 | 64.0 | 85.1 | 64.0 | 162.1 |
| | Contribution Ranking Pruning | **88.1** | **68.4** | **84.6** | **62.1** | **88.6** | **68.0** | **87.3** | **66.6** | **162.4** |
| D4 | Sequential Pruning | 80.6 | 62.1 | 77.6 | 55.0 | 81.9 | 60.3 | 82.8 | 60.3 | 210.7 |
| | Contribution Ranking Pruning | **84.5** | **65.9** | **82.3** | **58.9** | **84.8** | **64.1** | **84.1** | **63.0** | **214.1** |

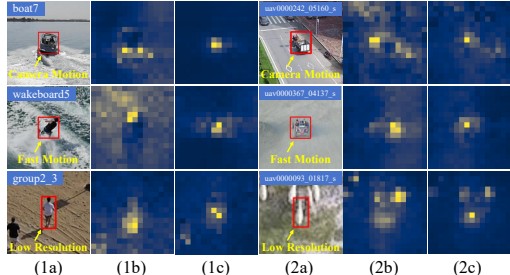
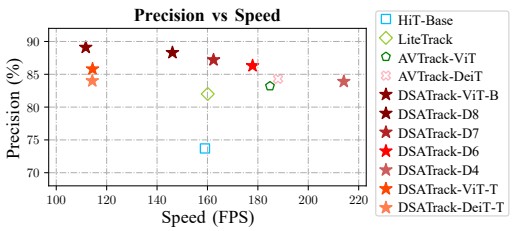

|     |     |     |     |     |     |
|-----|-----|-----|-----|-----|-----|
| (1a) | (1b) | (1c) | (2a) | (2b) | (2c) |

Figure 4: Visualization of correlation map (a) search region. (b) correlation map before the DSAM. (c) correlation map after the DSAM.

Figure 5: A comparison of speed and accuracy with state-of-the-art methods. The performance is the average across four datasets.

We evaluate our Dynamic Semantic-Aware Modeling (DSAM) against the Baseline and Static Semantic-Aware Modeling (SSAM). Dynamic semantic modeling refers to the dynamic semantic prediction method proposed in this paper, whereas static semantic modeling denotes the approach that applies a $3 \times 3$ convolution kernel to the correlation maps from the search region to the template and from the template to the search region, respectively. In this case, the receptive field remains fixed in shape. On DTB70, DSAM with ViT-B achieves 91.% precision and 70.6% success, outperforming both alternatives. Similar trends are observed on UAVDT and UAV123. Unlike SSAM's fixed-region modeling, DSAM adaptively captures semantic variability, enhancing tracking robustness. As shown in Figure 4, DSAM yields more focused correlation maps, effectively reducing background noise. These results confirm that DSAM significantly improves UAV tracking by modeling dynamic semantic correlations more accurately than static or baseline approaches.

### 4.4 Ablation Study on Pruning

Sequential Pruning (SP), similar to the proposed Contribution Ranking Pruning (CRP), removes layers 4 and 7 while retaining layer 10. However, SP performs pruning in a fixed sequential order, without explicitly evaluating the relative contribution of each layer to the overall tracking performance. In contrast, CRP introduces a hierarchical contribution ranking mechanism that quantifies each layer's importance before pruning, ensuring that critical layers are preserved while redundant ones are removed.

As shown in Table 5, CRP consistently outperforms SP across all backbones and datasets. For the D7 variant, CRP achieves higher precision and success rates on all four benchmarks (DTB70, UAVDT, VisDrone2018, and UAV123) with negligible changes in speed. The performance gap becomes even more pronounced in shallower configurations such as D4, where SP suffers from severe accuracy degradation due to over-pruning of essential layers, while CRP maintains stable accuracy with a similar inference speed (214.1 FPS vs. 210.7 FPS).

These results demonstrate that contribution-aware layer selection enables CRP to achieve a better balance between accuracy and computational efficiency. By adaptively ranking layers based on their hierarchical importance rather than following a rigid pruning schedule, CRP achieves a more robust pruning strategy, especially on challenging datasets such as VisDrone2018 and UAV123, where fine-grained spatial details are crucial for tracking performance.

## 5 Real Word UAV Test

As illustrated in Figure 6, we conduct a series of real-world UAV tracking experiments to further verify the practicality and deployment feasibility of the proposed DSATrack framework. Four representative tracking scenarios are presented, involving diverse targets such as a car, a motorbike, a dog, and a person, covering both rigid and non-rigid motion patterns as well as varying motion speeds and background complexities. These experiments were performed using a Jetson AGX Xavier edge computing platform to emulate realistic onboard conditions with limited computational resources.

To assess the runtime performance, we evaluate the inference speed of different DSATrack variants on the Xavier platform. The lightweight transformer-based variants, DSATrack-ViT-T and DSATrack-DeiT-T, achieve real-time tracking speeds of 27.6 FPS and 27.4 FPS, respectively, demonstrating their

suitability for low-latency applications on embedded hardware. In contrast, larger pruned variants such as DSATrack-D4 and DSATrack-D7 run at approximately 10 FPS, offering a stronger accuracy margin when higher computational capacity is available.

These results highlight the adaptability of DSATrack to diverse deployment scenarios. Depending on the trade-off between accuracy and efficiency, users can flexibly select the appropriate model variant to meet the demands of specific UAV task. The consistent performance across both dynamic and static scenes further confirms the robustness and scalability of the proposed framework in real-world UAV tracking applications.

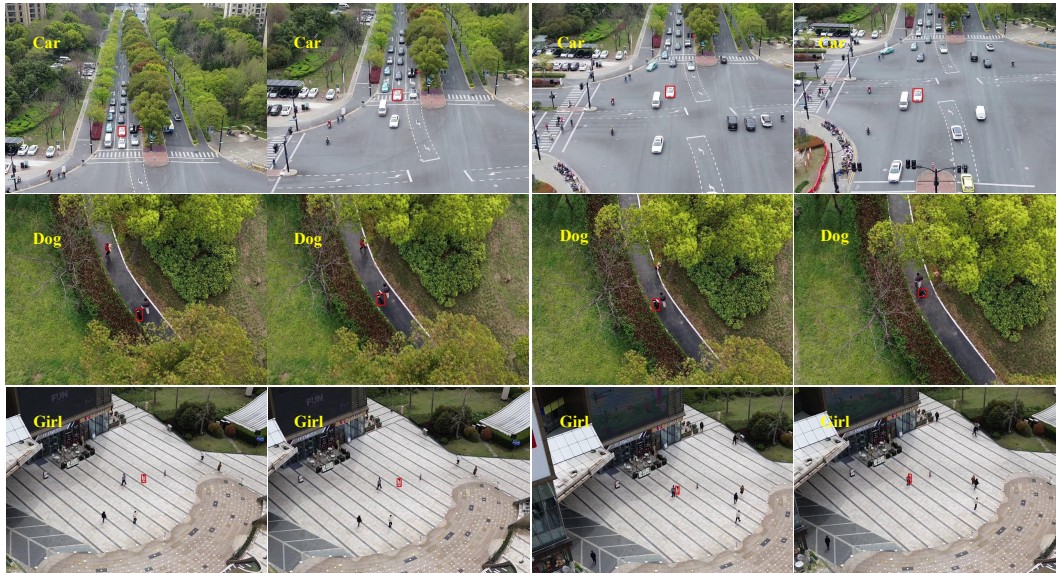

Figure 6: The real-world UAV tracking test includes tracking a car, motorcycle, dog, and person.

## 6    Conclusion and Limitation

The paper presents DSATrack, a dynamic semantic-aware UAV tracking framework addressing typical UAV Tracking challenges. It incorporates a Dynamic Semantic Relevance Generator to enhance feature fusion and localization, effectively reducing tracking drift. Additionally, a Hierarchical Contribution Ranking Pruning optimizes DSATrack by balancing tracking precision and real-time. Extensive experiments show that DSATrack achieves SOTA performance across various datasets, while ablation studies further validate the effectiveness. DSATrack also has certain limitations. Similar to most trackers, DSATrack is still a local tracker and lacks re-detection capability once the object is lost. In future work, we will explore global UAV tracking algorithms.

**Acknowledgement** This work was supported by National Natural Science Foundation of China (No.62576109, 62072112.

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

# A Technical Appendices and Supplementary Material

## A.1 Ablation Study on Correlation Modeling on attributes of UAV123.

Table 6: Ablation Study on Correlation Modeling on attributes of UAV123 including Camera Motion, Fast Motion and Low Resolution. Baseline refers to standard Transformer. SSAM represents the static semantic-aware modeling, while DSAM denotes the proposed dynamic semantic-aware modeling. The best results are highlighted in bold.

| Variant | Method | Camera Motion | | Fast Motion | | Low Resolution | | Overall | |
|---|---|---|---|---|---|---|---|---|---|
| | | Prec. | Succ. | Prec. | Succ. | Prec. | Succ. | Prec. | Succ. |
| DSATrack-ViT-B | Baseline | 90.9 | 69.9 | 84.0 | 64.2 | 74.2 | 50.5 | 87.3 | 67.2 |
| | Baseline+SSAM | 90.6 | 69.9 | 84.3 | 64.5 | 80.6 | 55.1 | 88.8 | 68.5 |
| | Baseline+DSAM | **93.5** | **72.2** | **89.8** | **68.6** | **81.8** | **55.9** | **90.2** | **69.4** |

As shown in Table 6, the ablation study demonstrates the effectiveness of Dynamic Semantic-Aware Modeling (DSAM) in enhancing tracking performance across various challenging attributes. Compared to the baseline Transformer, DSAM consistently improves both precision and success rates in all evaluated scenarios. Specifically, for DSATrack-ViT-B, DSAM outperforms the baseline by 2.6% and 2.3% in Camera Motion, 5.8% and 4.4% in Fast Motion, and 7.6% and 5.4% in Low Resolution. This suggests that DSAM effectively captures evolving target representations under complex UAV tracking conditions, where static modeling may struggle to adapt to rapid environmental changes. The performance gap is particularly evident in more challenging scenarios, such as Fast Motion and Low Resolution, where DSAM significantly enhances tracking accuracy and robustness.

To investigate the effect of different template input formats, we compare two variants under the ViT-D8 backbone: using only the template image (A Template) and combining the template with patch embeddings (Template&Patches). As shown in Table 8, incorporating patch-level information consistently improves performance across all UAV tracking benchmarks. Specifically, the Template&Patches design achieves the best overall results, with a precision/success gain of +1.6/+1.6 on UAVDT and +2.3/+1.7 on VisDrone2018 compared to the single-template input. These results demonstrate that introducing fine-grained patch features enhances the representation of template cues, leading to more robust matching and improved generalization across diverse UAV tracking scenarios

Table 7: Ablation study on the input format of the Template. The best results are highlighted in bold.

| Backbone | Method | DTB70 | | UAVDT | | VisDrone2018 | | UAV123 | |
|---|---|---|---|---|---|---|---|---|---|
| | | Prec. | Succ. | Prec. | Succ. | Prec. | Succ. | Prec. | Succ. |
| ViT-D8 | A Template | **89.5** | 69.3 | 85.0 | 61.8 | 88.5 | 67.4 | 86.4 | 65.7 |
| | Template&Patches | 89.3 | **69.5** | **85.2** | **63.4** | **90.8** | **69.1** | **87.6** | **67.0** |

## A.2 Ablation study on Token Elimination.

Table 8: Ablation study on token elimination. The best results are highlighted in bold.

| Backbone | Method | DTB70 | | UAVDT | | VisDrone2018 | | UAV123 | | FPS |
|---|---|---|---|---|---|---|---|---|---|---|
| | | Prec. | Succ. | Prec. | Succ. | Prec. | Succ. | Prec. | Succ. | |
| ViT-Base | w/o token elimination | 90.6 | 69.8 | 87.3 | 65.7 | 86.7 | 66.3 | 89.5 | 67.8 | 105.6 |
| | w/ token elimination | **91.2** | **70.6** | **88.1** | **66.3** | **87.1** | **67.0** | **90.2** | **69.4** | **111.7** |

To evaluate the effectiveness of token elimination, we conduct an ablation study across four UAV tracking benchmarks: DTB70, UAVDT, VisDrone2018, and UAV123. As shown in Table 8, incorporating token elimination consistently improves both tracking precision and success rate across all datasets, demonstrating the robustness and generalizability of the approach. For the ViT-Base backbone, token elimination yields consistent improvements over the baseline that retains all tokens. On DTB70, it achieves a gain of 0.6% in precision and 0.8% in success rate. Similar improvements

Table 9: Comparison with recent transformer-based SOT trackers. AVisT reports AUC/$OP_{50}$/$OP_{75}$; TrackingNet and LaSOT report AUC/$P_{\text{norm}}$/P; GOT-10k reports AO/$SR_{0.5}$/$SR_{0.75}$. "–" denotes not reported.

| Method | AVisT | | | TrackingNet | | | LaSOT | | | GOT-10k | | |
| | AUC | $OP_{50}$ | $OP_{75}$ | AUC | $P_{\text{norm}}$ | P | AUC | $P_{\text{norm}}$ | P | AO | $SR_{0.5}$ | $SR_{0.75}$ |
|---|---|---|---|---|---|---|---|---|---|---|---|---|
| SwinTrack | – | – | – | 81.1 | – | 78.4 | 67.2 | 70.8 | 47.6 | 71.3 | 81.9 | 64.5 |
| MixformerV2 | – | – | – | 83.4 | 88.1 | 81.6 | 70.6 | 80.8 | 76.2 | – | – | – |
| AsymTrack | – | – | – | 80.0 | 84.5 | 77.4 | 67.7 | 76.6 | 61.4 | 64.7 | 73.0 | 67.8 |
| LoRAT-B-224 | – | – | – | 83.5 | 87.9 | 82.1 | 71.7 | 80.9 | 77.3 | 72.1 | 81.8 | 70.7 |
| LoReTrack-256 | – | – | – | 82.9 | 81.4 | – | 70.3 | 76.2 | – | 73.5 | 84.0 | 70.4 |
| SeqTrack | 56.8 | 66.8 | 45.6 | 83.3 | 88.3 | 82.2 | 69.9 | 79.7 | 76.3 | 74.7 | 84.7 | 71.8 |
| AQATrack | – | – | – | 83.8 | 88.6 | **83.1** | **71.4** | **81.9** | **78.6** | 73.8 | 83.2 | 72.1 |
| OSTrack | 54.2 | 63.2 | 42.2 | 83.1 | 87.8 | 82.0 | 69.1 | 78.7 | 75.2 | 71.0 | 80.4 | 68.2 |
| **DSATrack-ViT-B (Ours)** | **60.2** | **69.1** | **50.2** | **84.1** | **88.6** | 82.7 | 69.4 | 78.6 | 74.6 | **75.0** | **85.6** | **73.7** |

Table 10: Comparison across four UAV tracking benchmarks and efficiency metrics. **Pre.**: precision; **Suc.**: success; FLOPs measured in GigaFLOPs (G).

| Method | DTB70 | | UAVDT | | VisDrone2018 | | UAV123 | | FLOPs(G) | Params(M) | Xavier(FPS) | RTX3090(FPS) |
| | Pre. | Suc. | Pre. | Suc. | Pre. | Suc. | Pre. | Suc. | | | | |
|---|---|---|---|---|---|---|---|---|---|---|---|---|
| HiT | 75.1 | 59.2 | 62.3 | 47.1 | 74.8 | 58.7 | 82.5 | 63.3 | 4.35 | 42.14 | 36.6 | 159.0 |
| AVTrack-ViT | 81.3 | 63.3 | 79.9 | 57.7 | 86.4 | 65.9 | 84.0 | 66.2 | 1.82 | 6.20 | 30.0 | 184.8 |
| LiteTrack | 82.5 | 63.9 | 81.6 | 59.3 | 79.7 | 61.4 | 84.2 | 65.9 | 12.81 | 49.59 | 15.9 | 160.1 |
| **DSATrack-D7** | **88.1** | **68.4** | **84.6** | **62.1** | **88.6** | **68.0** | **87.3** | **66.6** | 23.64 | 56.76 | 14.0 | 162.4 |
| DSATrack-D4 | 84.5 | 65.9 | 82.3 | 58.9 | 84.8 | 64.1 | 84.1 | 63.0 | 14.12 | 35.50 | 17.4 | **214.1** |
| DSATrack-ViT-T | 86.6 | 67.4 | 84.3 | 61.6 | 84.8 | 64.1 | 85.5 | 65.5 | 2.99 | 8.21 | 27.6 | 114.3 |
| DSATrack-DeiT-T | 84.5 | 65.7 | 85.0 | 62.0 | 82.0 | 60.8 | 84.4 | 64.5 | 2.99 | 8.21 | 27.4 | 114.4 |

are observed on UAVDT (+0.8 precision, +0.6 success), VisDrone2018 (+0.4 precision, +0.7 success), and UAV123 (+0.7 precision, +1.6 success). These gains are particularly significant under challenging scenarios such as camera motion, small object size, and background clutter, which are common in UAV tracking.

Beyond accuracy, token elimination also contributes to runtime efficiency. For instance, with the ViT-Base backbone, the frame rate increases from 105.6 FPS to 111.7 FPS, reflecting reduced computational overhead without sacrificing performance. This efficiency gain makes the model more suitable for real-time UAV deployment, especially on edge computing platforms with limited resources. Overall, the results validate that token elimination effectively filters out redundant or uninformative tokens, allowing the model to focus on semantically relevant features.

### A.3 Comparison with General Trackers

Table 9 presents a comprehensive comparison between DSATrack and recent transformer-based single-object trackers across four large-scale benchmarks: AVisT, TrackingNet, LaSOT, and GOT-10k. On AVisT, DSATrack-ViT-B achieves competitve overall performance, obtaining an AUC/OP50/OP75 of 60.2/69.1/50.2, demonstrating superior robustness under adverse visibility and occlusion. On Track-ingNet, DSATrack attains the highest AUC of 84.1 and $P_{\text{norm}}$ of 88.6, showing strong generalization to large-scale, high-diversity scenes. For LaSOT, DSATrack remains competitive with an AUC of 82.7 and precision of 74.6, comparable to the top-performing AQATrack while requiring no task-specific tuning. On GOT-10k, DSATrack delivers the best accuracy across all metrics (AO/SR0.5/SR0.75 = 75.0/85.6/73.7), surpassing previous transformer-based trackers such as OSTrack and LoRAT-B-224.

These results confirm that DSATrack maintains a favorable balance between tracking precision and robustness, matching state-of-the-art transformer trackers across both long-term and general-purpose benchmarks. Its consistent advantage on stricter metrics (OP75, SR0.75) highlights the model's improved localization accuracy and resilience in challenging conditions.

### A.4 Comparison with UAV Trackers

Table 10 compares DSATrack with recent transformer-based UAV trackers in terms of accuracy and efficiency across four representative UAV benchmarks—DTB70, UAVDT, VisDrone2018, and

UAV123—as well as computational complexity metrics. Among all competitors, DSATrack-D7 achieves the best overall accuracy, attaining the highest precision and success scores on every dataset (DTB70: 88.1/68.4, UAVDT: 84.6/62.1, VisDrone2018: 88.6/68.0, UAV123: 87.3/66.6). This indicates that the hierarchical contribution-based pruning and dynamic alignment strategy effectively preserve discriminative representations across diverse UAV scenarios.

Meanwhile, DSATrack-D4 maintains competitive accuracy while significantly improving efficiency, achieving the highest inference speed of 214.1 FPS on RTX 3090 and 17.4 FPS on NVIDIA Xavier, demonstrating excellent scalability for real-time embedded deployment. The lightweight variants DSATrack-ViT-T and DSATrack-DeiT-T further reduce parameters and FLOPs (8.21 M / 2.99 G) while sustaining robust tracking performance, highlighting the flexibility of the proposed framework in balancing accuracy and efficiency across different model scales and hardware settings.

Overall, DSATrack exhibits a favorable accuracy–efficiency trade-off, outperforming existing UAV-specific trackers in both precision and speed, thereby underscoring its practicality for onboard UAV perception applications.

## A.5 Broader Impact

The proposed UAV tracking method could benefit public safety and disaster monitoring by enabling real-time, robust tracking from aerial views. However, we also acknowledge the potential for misuse in surveillance applications and stress the importance of deploying such technology within clear legal and ethical boundaries.

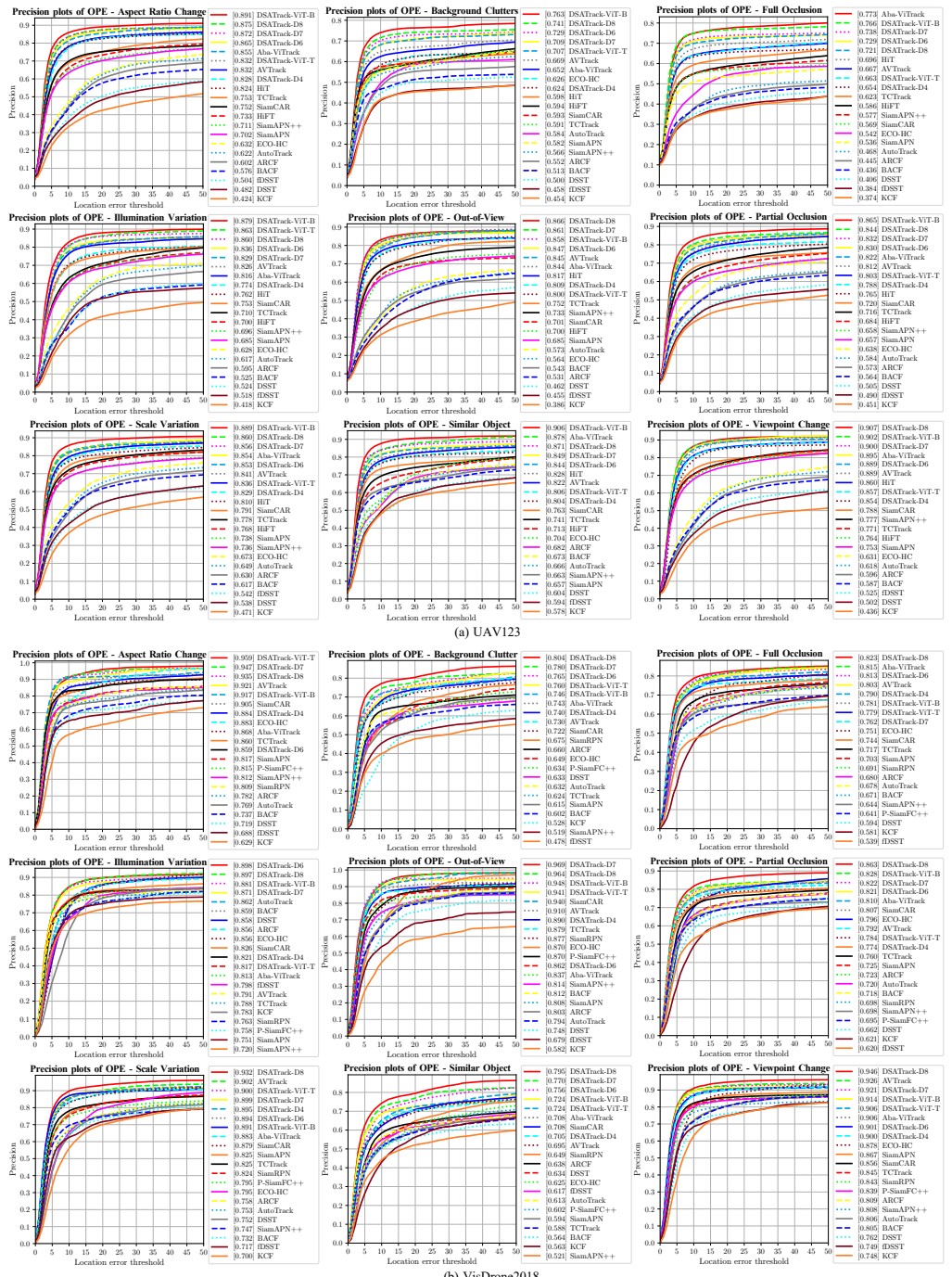

Figure 7: Comparison of state-of-the-art methods under comprehensive UAV challenges.

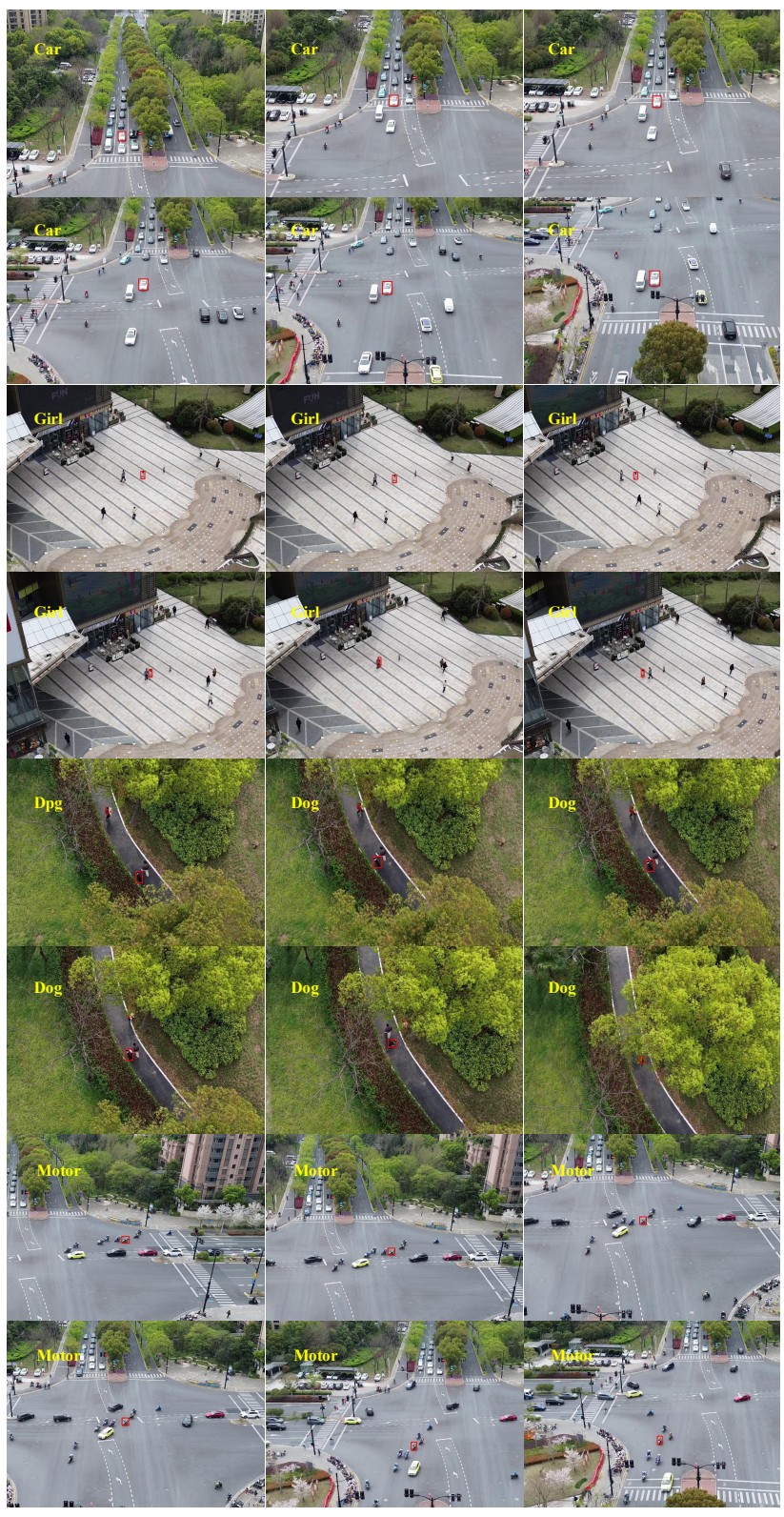

Figure 8: Real-world UAV tracking results on four representative scenarios, including car, motorcycle, dog, and person, demonstrating the adaptability of our method in diverse environments.

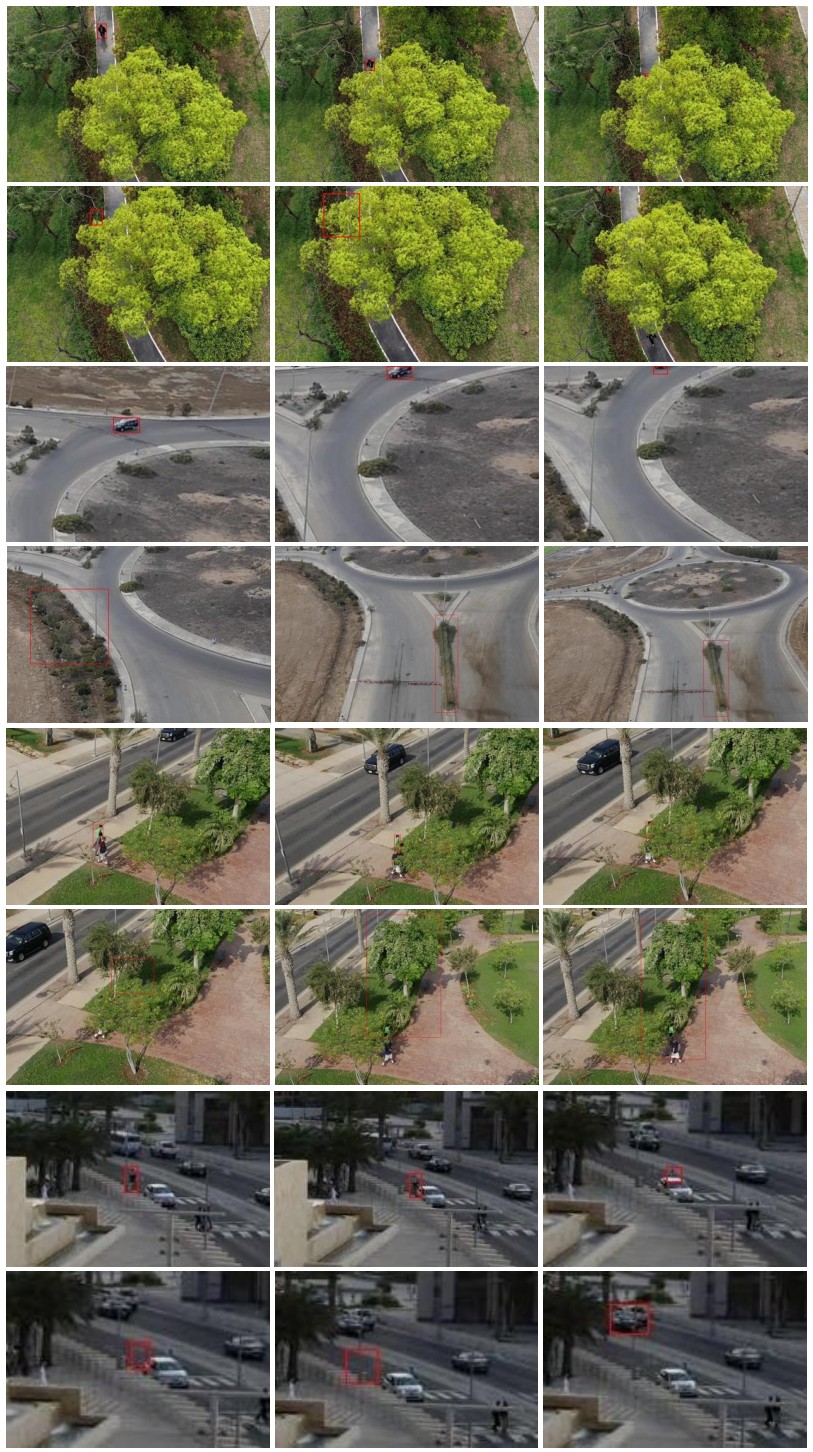

Figure 9: Real-world UAV tracking failure cases on four representative scenarios, including car, motorcycle, dog, and person, demonstrating the adaptability of our method in diverse environments.

