# OpenReview forum: "Dynamic Semantic-Aware Correlation Modeling for UAV Tracking"
_NeurIPS.cc/2025/Conference — NeurIPS 2025 poster_

### Official Review · Reviewer_YeVi · 2025-06-20

**Clarity:** 3
**Significance:** 3
**Originality:** 2
**Rating:** 5
**Confidence:** 5

**Summary:**

This work proposes a UAV tracker, named DSATrack, that utilizes semantic correlation to enhance tracking accuracy. The main contribution of DSATrack is the Dynamic Semantic Relevance Generator, which explores the semantic relevance according to the correlation map from the Transformer.
Benchmark experiments are comprehensive, including DTB70, UAVDT, VisDrone2018, and UAV123, and DSATrack reaches the SOTA performance. The ablation study on correlation modeling shows the effectiveness of the proposed Dynamic Semantic Relevance Generator.
The inference speed of the method is well considered by pruning, and the method shows efficiency in real-world tracking experiments.

**Questions:**

1. Will the authors provide the source code and trained weights for reproduction?
2. Will the authors provide the comparison on model size?
3. Will the authors plan to include larger or more benchmarks for evaluation?
4. How is model design intrinsically linked to the unique challenges of UAV tracking?
5. ​What fundamentally distinguishes the proposed Dynamic Semantic Relevance Generator from prior approaches to correlation modeling?
6. How will DSA generalize to datasets in common scenes, e.g., LaSOT?

**Ethical Concerns:**

["NO or VERY MINOR ethics concerns only"]

**Final Justification:**

The authors have addressed most of my concerns. It is recommended that the authors provide more explicit elaboration on the relevant issues in the main text. It is also suggested that the authors make further open-source contributions to the research community of UAV object tracking.​

**Limitations:**

Yes

**Paper Formatting Concerns:**

Minor adjustments are required to implement line breaks for subheadings in Section 3.1, ensuring adherence to the document's formatting conventions throughout the manuscript. For instance, Feature matching and Interaction.

**Quality:**

3

**Strengths And Weaknesses:**

Strength:

1. The experiments are exceptionally comprehensive and well-designed, featuring established UAV tracking datasets and including the presentation of multiple model variants with distinct configurations, including DTB70, UAVDT, VisDrone2018, and UAV123, and DSATrack.

2. Module design is reasonable and proven effective in experiments, especially in Table 4 and Table 5.

3. Several variants of the model design are proposed with different architectural settings for diverse demands of tracking.

4. The proposed DSATrack is deployed on an edge device for real-world application tests, where DSATrack-ViT-T and DSATrack-DeiT-T achieve 27.6 FPS and 27.4 FPS.

Weakness:

1. Although the evaluation benchmarks are well-established, their dataset scale appears insufficient by contemporary research standards.

2. Source code and model weights are not attached.

3. Comparison of model size between the proposed method and SOTA methods is not shown.

4. Limited novelty compared with baselines. The idea of correlation modeling has been studied in several works, including HiFT and TCTrack. The model architecture design is also based on RFGM and OSTrack.

5. Insufficient theoretical explanation of model design under UAV-specific scenarios. Special challenges of UAV tracking scenarios should be highlighted with cases.

6. The generalization performance of DSATrack on generic datasets was not addressed, e.g., LaSOT, GOT-10k.

---

> ### Author Rebuttal · Authors · 2025-07-31
>
> We sincerely thank you for your positive evaluation and constructive feedback. We are glad that you found our method and experimental analysis meaningful. In the following, we address your specific comments and suggestions, and describe how we plan to further improve the manuscript accordingly
> >***Q1**.  Although the evaluation benchmarks are well-established,How will DSA generalize to datasets in common scenes, e.g., LaSOT, GOT-10k?*
> - Thank you very much for your valuable suggestion. Our model is primarily designed for object tracking using UAVs, and we found your comment particularly insightful. Accordingly, we conducted additional evaluations on LaSO, GOT-10k, TrackingNet, and AvisT, and the results demonstrate that our approach achieves competitive performance compared to mainstream single-object tracking methods.
> |Method|---|AVisT|---|---|TrackingNet|---|---|LaSOT|---|---|GOT-10k|---|
> |-|-|-|-|-|-|-|-|-|-|-|-|-|
> ||$AUC$|$OP_{50}$|$OP_{75}$|$AUC$|$P_{norm}$|$P$|$AUC$|$P_{norm}$|$P$|$AO$|$SR_{0.5}$|$SR_{0.75}$|
> |SwinTrack|-|-|-|81.1|-|78.4|67.2|70.8|47.6|71.3|81.9|64.5|
> |MixformerV2|-|-|-|83.4|88.1|81.6|70.6|80.8|76.2|-|-|-|
> | AsymTrack |-|-|-|80.0 |84.5| 77.4 |67.7| 76.6| 61.4| 64.7| 73.0| 67.8 |
> |LoRAT-B-224|-|-|-|83.5|87.9|82.1| 71.7| 80.9| 77.3|72.1|81.8|70.7|
> |LoReTrack-256|-|-|-|82.9| 81.4|-|70.3| 76.2| -|73.5|84.0| 70.4|
> |SeqTrack|56.8|66.8|45.6|83.3|88.3|82.2|69.9|79.7|76.3|74.7|84.7|71.8 |
> |AQATrack|-|-|-|83.8|88.6|**83.1**|**71.4**|**81.9**|**78.6**|73.8|83.2|72.1|
> |OSTrack|54.2|63.2|42.2| 83.1|87.8|82.0|69.1|78.7|75.2|71.0|80.4|68.2|
> |**DSATrack-ViT-B**|**60.2**|**69.1**|**50.2**|**84.1**|**88.6**|82.7|69.4|78.6|74.6|**75.0**|**85.6**|**73.7**|
>
>
> - **We will include all these experimental results in the revised version during the camera-ready stage**.
>
> >***Q2**. Will the authors provide the source code and trained weights for reproduction?*
> - Thank you very much for your question. **In fact, after receiving your comment, we promptly organized the code and model weights. However, we later noted that NeurIPS does not allow anonymous links this year.** If our paper is accepted, **we will publicly release all training and testing code, along with the model weights, to ensure transparency and reproducibility**.
>
>
> >***Q3**. Will the authors provide the comparison on model size?*
> - Thank you for your valuable suggestion. We have provided the model size and inference speed as a footnote for your reference. Our approach achieves a balance between speed and accuracy, allowing different model sizes to be selected based on the computational platform for performing the tracking task.
>
> |Method|DTB70-Pre.|DTB70-Suc.|UAVDT-Pre.|UAVDT-Suc.|VisDrone-Pre.|VisDrone-Suc.|UAV123-Pre.|UAV123-Suc.|Flops(G)|Params(M)|Xavier(FPS)|RTX3090(FPS)|
> |-|-|-|-|-|-|-|-|-|-|-|-|-|
> |HiT|75.1|59.2|62.3|47.1|74.8|58.7|82.5|63.3|4.35|42.14|36.6|159.0|
> |AVTrack-ViT|81.3|63.3|79.9|57.7|86.4|65.9|84.0|66.2|1.82|6.2|30.0|184.8|
> |LiteTrack|82.5|63.9|81.6|59.3|79.7|61.4|84.2|65.9|12.81|49.59|15.9|160.1|
> |**DSATrack-D7**|**88.1**|**68.4**|84.6|**62.1**|**88.6**|**68.0**|**87.3**|**66.6**|23.64|56.76|14.0|162.4|
> |DSATrack-D4|84.5|65.9|82.3|58.9|84.8|64.1|84.1|63.0|14.12|35.5|17.4|**214.1**|
> |DSATrack-ViT-T|86.6|67.4|84.3|61.6|84.8|64.1|85.5|65.5|2.99|8.21|27.6|114.3|
> |DSATrack-DeiT-T|84.5|65.7|**85.0**|62.0|82.0|60.8|84.4|64.5|2.99|8.21|27.4|114.4|
>
> - **We will include this information in the paper, as it greatly contributes to improving the quality of our manuscript.**
>
> >***Q4**. The idea of correlation modeling has been studied in several works, including HiFT and TCTrack. The model architecture design is also based on RFGM and OSTrack.*
> - Thank you very much for your comments and questions. **Our method differs significantly from HIFT and TCTrack. Both HIFT and TCTrack are based on CNN architectures, where the correlation map is generated through convolutional operations**. Specifically, they perform matching between the template and search region in a holistic manner using convolution.
>
> - **In contrast, our method adopts a Transformer-based correlation mechanism, where matching is conducted in a point-to-point manner based on token-level similarity**. This point-wise matching inherently **lacks relational information between different positions and does not capture the global structure of the target**, which often leads to ambiguous matching.
>
> - **To address this limitation and enhance the semantic perception capability of the Transformer, we introduce a Semantic-Aware Generator that constructs a semantic relationship**. This relationship adaptively identifies which points in the template are semantically related to which regions in the search area, enabling more effective correlation fusion and significantly alleviating the semantic ambiguity problem of Transformers.
>
> - Moreover, since Transformer computes similarity based on token-wise interactions, it requires a richer set of representative semantic tokens to achieve accurate correlation. To this end, we employ a Semantic-Aware Correlation Map to guide the selection of semantic tokens, thereby improving the semantic expressiveness of the template. The enhanced semantic tokens in turn lead to a more accurate correlation map. **These two components — semantic tokens and semantic correlations — mutually reinforce and benefit each other**.
>
> - Compared with OSTrack and RFGM, our framework also adopts a one-stream Transformer architecture, which has become a strong and widely adopted baseline. However, unlike existing works, our method explicitly focuses on addressing the semantic modeling issue, which remains a core limitation of one-stream Transformers and also forms the motivation of our work.
>
> - We have further provided a comparative analysis of the four methods for your reference, as shown below:
> |Method|Hift|TCTrack|**DSATrack**|RFGM|OSTrack|
> |-|-|-|-|-|-|
> |**Architecture**|CNN-based Correlation|CNN-based Correlation|One Stream Transformer|One Stream Transformer|One Stream Transformer|
> |**Correlation Method**|Convolutional Operation|Convolutional Operation|Dynamic Semantic Matching|Transformer-based Matching|Transformer-based Matching|
> |**Semantic Aware Method**|Correlation Refinement by Transformer|Correlation Refinement by Transformer|Generate a dynamic semantic relationship|Point-to-Point|Point-to-Point|
> |**Semantic Tokens**|No|No|With Dynamic Semantic Tokens|No|No|
> |**Mutually enhance**|Only Refinement|Only Refinement|Mutually enhance between Semantic Tokens and Correlation|No|No|
>
> >***Q5**. How is model design intrinsically linked to the unique challenges of UAV tracking? Special challenges of UAV tracking scenarios should be highlighted with cases.*
> - Thank you for your comments and questions. **As mentioned in Lines 28–33 of the Introduction, our method specifically targets the challenges commonly encountered in UAV tracking scenarios, such as camera motion, fast motion, and low resolution**. These scenarios often lead to motion blur or the loss of semantic information, which significantly degrade the performance of existing trackers. However, current one-stream Transformer-based UAV tracking models do not explicitly address these issues.
>
> - To this end, we propose a semantic-aware approach to enhance the model’s generalization ability under such challenging conditions. **As illustrated in Figure 1, our method demonstrates clear advantages over existing CNN-based correlation and Transformer-based correlation strategies**.
>
> - Furthermore, in Figure 3, we present a qualitative comparison under the camera motion, fast motion, and low-resolution scenarios with other state-of-the-art trackers. For your convenience, we have also summarized the results of Figure 3 into a table, which further supports the effectiveness of our proposed semantic-aware design in mitigating the ambiguity problem caused by point-wise matching in Transformers.
>  |**Method**|Low Resolution|Fast Motion|Camera Motion|
> |-|-|-|-|
> |**ECO-HC**|54.7|67.9|80.0||
> |**SiamAPN++**|58.8|75.9|70.8|
> |**BACF**|59.8|57.3|77.3|
> |**KCF**|64.2|56.9|70.0|
> |**TCTrack**|64.4|73.8|78.5|
> |**AVTrack**|73.8|82.7|86.5|
> |**DSATrack-ViT-B**|74.5|74.5|85.8|78.1|
> |**DSATrack-ViT-T**|77.1|76.5|86.3|
> |**DSATrack-D8**|86.8|**88.4**|**89.7**|
> |**DSATrack-D7**|**87.2**|82.0|87.0|
> |**DSATrack-D6**|76.2|85.9|87.7|
> |**DSATrack-D4**|80.6|86.0|87.6|
>
> - In addition, we conduct an ablation study in Appendix A.1 (Table 6), where our semantic-aware method achieves performance improvements of 2% to 3% under the three aforementioned challenges. For your reference, we include the table below:
> |Method| DTB70-Prec. | DTB70-Succ.|UAVDT-Prec.|UAVDT-Succ.|VisDrone2018-Prec.|VisDrone2018-Succ.|UAV123-Prec.|UAV123-Succ.|
> |-|-|-|-|-|-|-|-|-|
> |Baseline|87.7|67.9|85.0|63.3|86.5|65.7|87.3|67.2|
> |Baseline+DSAM|**91.2**|**70.6**|**88.1**|**66.3**|87.1|**67.0**|**90.2**|**69.4**|
>
> - **Thank you very much for your valuable suggestion. We will further enhance the explanation of the addressed challenges in our revision, as this greatly contributes to improving the quality of our manuscript.**

---

> > ### Comment · Reviewer_YeVi · 2025-08-01
> >
> > I would like to thank the authors for addressing my concerns. I decided to raise the score to 5.

---

> > > ### Author Response · Authors · 2025-08-01
> > >
> > > Thank you very much for your positive recognition of our work. We are sincerely grateful that you raised your score to a 5—this is highly encouraging and means a great deal to us.

---

### Official Review · Reviewer_AQ2d · 2025-06-27

**Clarity:** 3
**Significance:** 3
**Originality:** 3
**Rating:** 4
**Confidence:** 3

**Summary:**

The paper addresses the problem of tracking objects with the use of a drone equipped with a camera. The proposed solution is based on the use of a transformer architecture. The specific contribution is related to the addition of a semantic component to improve tracking ability and a pruning approach in the transformer blocks to reduce complexity and increase tracking speed.
This reviewer found the idea of adding a semantic component to better track the object in the scene relevant. It is based on generating template semantic tokens and then computing a correlation with the scene/search tokens.  This allows the implementation of a dynamic semantic relevance generator.

The paper offers numerical results using various datasets and a comparison with other architectures used for tracking. Additionally, some real experimentation is also documented.

**Questions:**

In addition to the above comments:

- Can you elaborate more on how the template tokens are generated?

- The Semantic Aware Modeling section might be integrated in the previous one, since essentially this describes just a refinement of the correlation map.

- Would it be possible to run experiments that show the tracking ability of multiple objects?

- The section about real-world experiments does not clarify how the initialization of the tracking process is done. That is, how the object is selected by correctly positioning the bounding box.  Additionally, the reported results show "qualitative" performance (some frames). Would it be possible to define and report some quantitative performance metrics?

- A final minor remark is that the terminology "UAV tracking" is ambiguous since at first it may be interpreted as tracking the location of the UAV and not using a UAV equipped with a camera to monitor and track objects in a scene.

**Ethical Concerns:**

["NO or VERY MINOR ethics concerns only"]

**Final Justification:**

I appreciated the reply and additional experimental results made in the rebuttal phase. I have increased the significance score.

**Limitations:**

Yes

**Quality:**

3

**Strengths And Weaknesses:**

The proposed approach has been proven to offer the best performance and even the second-best performance with lighter versions.
The idea of adding a semantic scene component is interesting. What is not fully understood from the paper is how the template semantic tokens are generated. It is assumed that this is achieved by pre-training with a number of possible known objects and scenes in a supervised fashion. If so, the architecture is limited to tracking "known" objects.

Additionally, tracking is herein limited to a single object. Would it be possible to simultaneously track multiple objects also belonging to different classes?

And what about using multiple cameras in a multi-drone setup? Is it something to be considered and possibly implemented by extending the current architectures?

---

> ### Author Rebuttal · Authors · 2025-07-30
>
> We sincerely appreciate your time and thoughtful review. We would like to take this opportunity to clarify the core ideas, address the raised issues in detail, and provide additional experimental results to support the effectiveness of our method.
>
> >***Q1**.  The idea of adding a semantic scene component is interesting. What is not fully understood from the paper is how the template semantic tokens are generated.*
> - Thank you for your insightful question. It precisely targets the core innovation of our paper. I am genuinely excited to share our idea with you.
>
> - Our method does not require prior knowledge of the target's category. Instead, our supervision only indicates where an object is present, without specifying what the object is. Given this limited information, how can we perform semantic modeling?
>
> - We address this challenge by introducing a token dropping strategy on the template. Specifically, we enforce the model to discard tokens with low semantic relevance and retain those with richer semantic content. To achieve this in an end-to-end manner, we employ the Gumbel-Softmax[1], which allows us to learn discrete token selection during training. Tokens with lower scores—i.e., less semantic information—are dropped, while higher-scoring tokens are preserved. The formulation of Gumbel-softmax[1] is as bellow:
> - $\mathcal{M} = \text{Gumbel-Softmax}\left( \log\text{-}\text{Softmax}(\text{MLP}(C'_{zx})) \right).$
> - The  Correlation Map $C'_{zx}$ is fed into an MLP and Gumbel-softmax to evaluate the importance of the Template Tokens.
> Less semantic tokens are discarded based on $\mathcal{M} \in \{0,1\} \in \mathbb{R}^{N_z \times N_z}$ to accelerate computation.
> - This training strategy enables our model to acquire the ability to identify semantic tokens even during inference.
> As a result, the patch memory progressively stores an increasing number of semantic tokens, which in turn enhances the quality of the semantic-aware correlation map during matching.
> This improved correlation map further facilitates more accurate selection of semantic tokens, forming a mutually reinforcing cycle that continuously strengthens both components.
>
>
>
> >***Q2**. It is assumed that this is achieved by pre-training with a number of possible known objects and scenes in a supervised fashion. If so, the architecture is limited to tracking "known" objects.*
> - As indicated in Q1, our semantic token selection process does not require any prior knowledge of the object category.
> Instead, we leverage Gumbel-softmax to enforce the model to learn, in a self-supervised manner, which tokens correspond to potential targets.
> Therefore, our method is inherently category-agnostic—it operates without any need to know what the target is in advance.
>
> >***Q3**. Additionally, tracking is herein limited to a single object. Would it be possible to simultaneously track multiple objects also belonging to different classes?*
> - **Thank you for your valuable suggestion. Although our original focus was on single object tracking, following your advice, we conducted additional experiments on multi-object tracking during the rebuttal period.**
> - Specifically, we crop multiple search regions and corresponding templates for each target within a frame. During inference, different targets are processed simultaneously by stacking them along the batch dimension, allowing our model to track multiple objects in parallel. The dataset used for this experiment is LaGOT，which is a multiple object tracking dataset. The experimental results are as follows：
> | Metric| DSATrack-D8 | SuperDiMP | DSATrack-D7 | PrDiMP-50 | DSATrack-D6 | PrDiMP-18 | DiMP-50 | DiMP-18 | QDTrack | OVTrack |
> |-|-------------|-----------|-------------|-----------|-------------|-----------|---------|---------|---------|---------|
> | **Success**    | 60.1        | 58.7      | 58.4        | 55.9      | 56.8        | 53.3      | 53.2    | 50.8    | 19.4    | 13.6    |
> | **Precision**  | 56.7       | 56.6     | 55.1       | 53.9     | 53.3       | 51.8     | 51.5   | 48.7   | 18.5   | 12.6   |
>
> >***Q4**. And what about using multiple cameras in a multi-drone setup? Is it something to be considered and possibly implemented by extending the current architectures?*
> **Thank you for the thoughtful question. Extending single-drone tracking systems to multi-camera and multi-drone setups is indeed a meaningful and challenging direction. Such an extension would require addressing several key issues, including**:
>
> - **Cross-view calibration and synchronization** to align observations from different drones;
>
> - **Inter-drone communication and coordination mechanisms** for consistent identity management and data association;
>
> - **Robust multi-view fusion strategies** to combine information from different viewpoints, especially under occlusion or viewpoint variation.
> - **In future work, we will make efforts to extend our approach to multi-camera and multi-drone systems as per your suggestion. We believe this would lead to a highly impactful and challenging research direction**.
>
> >***Q5**.Can you elaborate more on how the template tokens are generated?*
> - Thank you for your comment. The design is based on RFGM[2], we mentioned this part in Section 3.1, line 102 of the paper.   Each template $z_t \in \mathbb{R}^{3 \times H \times W} $ is divided into non-overlapping patches of size $S \times S $, producing $N_z = \frac{H \times W}{S^2}$ patches. The patches are projected into token embeddings via a shared convolutional layer:
> $$
> T^z_t = \{ T^z_1, T^z_2, ..., T^z_{N_z} \}, \quad T^z_i \in \mathbb{R}^C
> $$
> where $C$ is the token embedding dimension. These tokens are appended to the token memory $\mathcal{M}_{t-1}$ from previous frames to form a candidate token pool:
> $$
> \mathcal{M}\_{t} = \mathcal{M}\_{t-1} \cup T^z\_t
> $$
>
> >***Q6**. The Semantic Aware Modeling section might be integrated in the previous one, since essentially this describes just a refinement of the correlation map.*
> - Thank you for the suggestion. We agree with your comment and will integrate the Semantic-Aware Modeling section into the previous part in the revised version.
>
> >***Q7**. That is, how the object is selected by correctly positioning the bounding box.*
> - At the beginning, the UAV hovers steadily in the air. The operator selects the initial bounding box from the live video feed, and the coordinates are sent back to the onboard Jetson AGX Xavier to initialize the tracker. The tracking process then begins.
>
>
> >***Q8**. Would it be possible to define and report some quantitative performance metrics?*
> - Thank you for your valuable suggestion. During the rebuttal period, we manually annotated the four recorded videos and conducted quantitative evaluations. The results are shown below.
> - We greatly appreciate your recommendation and will include this performance table in the camera-ready version. The suggestion has been very helpful in improving the quality of our manuscript.
> | Model             |   AUC |   Precision |   Norm Precision |
> |:-|------:|------------:|-----------------:|
> | DSATrack-ViT-B  | 66.9 |82.4  |79.7 |
> | DSATrack-D8  | 66.4 |83.7  |81.9 |
> | DSATrack-D7  | 61.6 |76.2  |72.0 |
> | DSATrack-D4  | 61.8  |  78.3|74.9  |
> | AVTrack  | 62.0 |   76.3  | 74.2|
>
>
>
> >***Q9**. A final minor remark is that the terminology "UAV tracking" is ambiguous since at first it may be interpreted as tracking the location of the UAV and not using a UAV equipped with a camera to monitor and track objects in a scene.*
> - Thank you for your valuable suggestion. We agree that the term “UAV tracking” may be ambiguous. In the revision, we will replace it with “Object Tracking with UAV” to more accurately convey that the task involves using UAVs to track objects in the scene.
>
> [1]. Jang E, Gu S, Poole B. Categorical reparameterization with gumbel-softmax[J]. arXiv preprint arXiv:1611.01144, 2016.

---

> > ### Comment · Reviewer_AQ2d · 2025-08-05
> >
> > I'd like to thank the authors for their clarifications. I also appreciated the additional experimental results. I will increase the significance score a bit.

---

> > > ### Author Response · Authors · 2025-08-05
> > >
> > > We sincerely thank the reviewer for the positive recommendation and for acknowledging the value of our clarifications and additional experiments.
> > >
> > > We appreciate the effort to reassess the significance of our work and improve the score, and we are encouraged that the improvements helped to better convey the contribution and impact of our method.
> > >
> > > Thank you again for your thoughtful and supportive review.

---

### Official Review · Reviewer_cKGP · 2025-06-27

**Clarity:** 3
**Significance:** 2
**Originality:** 3
**Rating:** 4
**Confidence:** 4

**Summary:**

This paper proposes a UAV tracking framework called DSATrack, which aims to address the lack of semantic awareness in existing methods. DSATrack introduces a dynamic semantic correlation generator and combines this generator with a Transformer-based correlation graph to explore semantic relations. This design enhances the ability to extract key information from the template within the search region. The framework also includes a pruning technique, thus achieving a balance between speed and accuracy. The experimental results on multiple datasets demonstrate its effectiveness and robustness.

**Questions:**

1.The paper states that the inputs of DSATrack are the template and the patch. It is unclear where the patch comes from. It is also unclear how the model performs without the patch. This should be explained and tested in an ablation study.
2.The description of the Dynamic Semantic-aware Transformer in Figure 2 is confusing. It is unclear if Semantic-aware Correlation Map is generated by Semantic Aware Modeling module or if it is unrelated to it, as Figure 2 suggests. The authors should review Figure 2 and the text and decide if they need to revise the figure or the description. Figure 2 also shows both a Dynamic Semantic-aware Transformer block and a Transformer block. It is unclear which Transformer layers are removed.
3.In Table 3, LiteTrack is not left aligned.
4.The paper does not explain the difference between SSAM and DSAM.
5.I also doubt the efficiency and accuracy of the DSATrack-ViT-B model.

**Ethical Concerns:**

["NO or VERY MINOR ethics concerns only"]

**Final Justification:**

Thanks for the detailed response. I tend to raise the rating, as my concerns have been mostly resolved.

**Limitations:**

1. Recent advanced SOT transformer-based methods are missing in the experiments, such as OSTrack, LoReTrack, LoRAT, MixformerV2, AQATrack, AsymTrack. Although not UAV-specific trackers, it is necessary to compare the proposed tracker with these non-real-time SOTA trackers for a comprehensive evaluation.
2. Discussion on Computational Complexity, while DSATrack outperforms SOTA models in speed, the paper lacks a detailed comparison of FLOPs and parameter.
3. Failure Case Analysis: It is recommended to include a detailed failure analysis, highlighting scenarios where DSATrack struggles.

**Quality:**

2

**Strengths And Weaknesses:**

This paper brings dynamic semantic aware correlation modeling into the field of aerial tracking. It has good validity. However, the description of the method can be confusing. Some concepts are not clearly explained. The experiments are not thorough enough. In addition, the credibility of the efficiency and accuracy of the DSATrack-ViT-B model is open to question.

---

> ### Author Rebuttal · Authors · 2025-07-30
>
> **We sincerely apologize for the omission. In fact, many of the experiments you pointed out have already been conducted in accordance with the suggestions**. However, **due to NeurIPS’s strict 9-page limit for single-column submissions— which provides even less space than the standard 8-page double-column**— we were unfortunately forced to leave out some of the content.
>
> **In camera ready,  NeurIPS allows an extension to 10 pages, and we will make sure to include all the experiments and address your comments directly in the main text. We truly hope you will reconsider our submission**.
>
> >***Q1**. I also doubt the efficiency and accuracy of the DSATrack-ViT-B model.*
> - The discrepancy from previous results arises because our method is developed based on the codebase of GRM (CVPR 2023)[1], in which the parameter **exclude_invalid_frames** of testing code is set to **True** by default.
> - **Through multiple rounds of testing, we found that enabling this setting to True to yield higher performance**. However, after reviewing relevant literature and consulting with peers, we found that there is **no established consensus** on whether invalid frames should be excluded.
> - **To ensure the rigor of our evaluation**, we reset this parameter to **False**. Our model underwent **extensive training and repeated evaluation**, and the results presented in the paper reflect the most rigorous setting.
> - **We will release the complete training and evaluation code, along with model weights, for full transparency and reproducibility**.
>
> >***Q2**. It is unclear which Transformer layers are removed.*
> - In Section 3.3 of the paper, Hierarchical Contribution Ranking Pruning is used to prune Transformer blocks. **Table 1 of the paper presents the contribution scores of each layer (See Line 179-180)，I have restated the data from Table 1 below for your reference**:
> |Dataset|Layer2|Layer3|Layer4|Layer5|Layer6|Layer7|Layer8|Layer9|Layer10|Layer11|Layer12|
> |-|-|-|-|-|-|-|-|-|-|-|-|
> |UAV123|0.1325|0.0739|0.0615|0.0581|0.0429|0.0457|0.0438|0.0358|0.0428|0.0657|0.1444|
>
> - **While Table 2 of the paper lists the specific layers that are pruned (See Line 179-180), I have restated the data from Table 2 below for your reference**:
> |Variant|RemovedLayers|Variant|RemovedLayers|
> |-|-|-|-|
> |DSATrack-D8|6,7,9,10|DSATrack-D6|5,6,7,8,9,10|
> |DSATrack-D7|6,7,8,9,10|DSATrack-D4|3,5,6,7,8,9,10,11|
>
>
> >***Q3**. The paper states that the inputs of DSATrack are the template and the patch. It is unclear where the patch... It is..., should be explained and tested in an ablation study.*
> - Thank you for your comment. We apologize for the lack of clarity regarding the “patch”. **It is not our main contribution**. The design is based on RFGM[2], we mentioned this part in Section 3.1, line 102 of the paper.   Each template $z_t \in \mathbb{R}^{3 \times H \times W} $ is divided into non-overlapping patches of size $S \times S $, producing $N_z = \frac{H \times W}{S^2}$ patches. The patches are projected into token embeddings via a shared convolutional layer:
> $$
> T^z_t = \{ T^z_1, T^z_2, ..., T^z_{N_z} \}, \quad T^z_i \in \mathbb{R}^C
> $$
> where $C$ is the token embedding dimension. These tokens are appended to the token memory $\mathcal{M}_{t-1}$ from previous frames to form a candidate token pool:
> $$
> \mathcal{M}\_{t} = \mathcal{M}\_{t-1} \cup T^z\_t
> $$
> - The token update mechanism can be referred to in RFGM[2]. **Thank you for your suggestion — we will provide a detailed explanation of the implementation in the appendix of the camera-ready version**.
>
> - The ablation study is as below:
> |Backbone|Method|DTB70-Prec.|DTB70-Succ.|UAVDT-Prec.|UAVDT-Succ.|VisDrone-Prec.|VisDrone-Succ.|UAV123-Prec.|UAV123-Succ.|
> |-|-|-|-|-|-|-|-|-|-|
> |ViT-D8|One template|89.5|69.3|85.0|61.8|88.5|67.4|86.4|65.7|
> |ViT-D8|Temaple&Patches|89.3|69.5|85.7|63.4|90.8|69.1|87.6|67.0|
>
> >***Q4**. The description of the Dynamic Semantic-aware Transformer in Figure 2 is confusing. It is unclear...*
> - **The Semantic aware Correlation Map is outputed by Semantic-Aware Modeling. In addition, we will enclose the Graph and Semantic Model label within a single box** in Figure 2 to present more clearly.
> - In the main pipeline (top of Figure 2), the Graph block  **“Semantic-Aware Modeling”** outputs a tensor labeled with **“Semantic-aware Correlation Map”** that follows the solid arrow into the **Hybrid Attention**.
> - As noted in lines 109–111 of the manuscript,
>  > “Using the Semantic Aware Modeling, we fuse the similarities of features with similar semantics to obtain an optimized Semantic-Aware Correlation Map.”  (see  Line109-111).
> * **Equation (7)** (line 140-142) – We will incoparate the description :
>  > “The Semantic aware Correlation Map is obtained through Semantic-Aware Modeling.”
>   $$
>   C'\_{zx} = \Lambda^{-\tfrac12}\,\hat{E}\,\Lambda^{-\tfrac12}\,C\_{zx}^{\top}\,W\_v ,
>   $$
>   where $C'\_{zx}$ is precisely the **Semantic-aware Correlation Map**.
> - **Thank you for your suggestion, whichi is helpful to improve our manuscript**.
>
> >***Q5**. In Table 3, LiteTrack is not left aligned.*
> - Thank you for your reminder. I will incorporate the revision, it is very helpful for enhancing the quality of the manuscript.
>
> >***Q6**. The paper does not explain the difference between SSAM and DSAM.*
> - SSAM employs convolution operation. The receptive field is limited to the size of the convolutional kernel. It can only capture and fuse semantic correlations within a local region. In contrast, our proposed DSAM enables the model to adaptively integrate the relevant semantic information across global semantic correlation.
> - Our initial design aimed to capture semantic correlations using 4D convolution (SSAM), which indeed brought performance improvements. Motivated by these gains, we further extended the idea by exploring whether dynamically selected (DSAM) could lead to even greater enhancements. The experimental results confirmed the validity of this hypothesis. As shown in the table, our method achieved improvements across all datasets, with gains of 2 to 3%.
> |Method| DTB70-Prec. | DTB70-Succ.|UAVDT-Prec.|UAVDT-Succ.|VisDrone2018-Prec.|VisDrone2018-Succ.|UAV123-Prec.|UAV123-Succ.|
> |-|-|-|-|-|-|-|-|-|
> |Baseline|87.7|67.9|85.0|63.3|86.5|65.7|87.3|67.2|
> |Baseline+SSAM|88.0|68.3|85.5|62.6|**87.8**|66.6|88.8|68.5|
> |Baseline+DSAM|**91.2**|**70.6**|**88.1**|**66.3**|87.1|**67.0**|**90.2**|**69.4**|
>
> >***Q7**. Recent advanced SOT transformer-based methods are missing in the experiments, such as OSTrack, LoReTrack, LoRAT, MixformerV2, AQATrack...Although not UAV-specific trackers*
> - We sincerely appreciate your suggestions. In comparative analysis, DSATrack was evaluated against several sota methods. The results demonstrate that DSATrack achieves competitive performance compared to advanced tracking algorithms while attaining the high speed.
> |Method|---|AVisT|---|---|TrackingNet|---|---|LaSOT|---|---|GOT-10k|---|
> |-|-|-|-|-|-|-|-|-|-|-|-|-|
> ||$AUC$|$OP_{50}$|$OP_{75}$|$AUC$|$P_{norm}$|$P$|$AUC$|$P_{norm}$|$P$|$AO$|$SR_{0.5}$|$SR_{0.75}$|
> |SwinTrack|-|-|-|81.1|-|78.4|67.2|70.8|47.6|71.3|81.9|64.5|
> |MixformerV2|-|-|-|83.4|88.1|81.6|70.6|80.8|76.2|-|-|-|
> | AsymTrack |-|-|-|80.0 |84.5| 77.4 |67.7| 76.6| 61.4| 64.7| 73.0| 67.8 |
> |LoRAT-B-224|-|-|-|83.5|87.9|82.1| 71.7| 80.9| 77.3|72.1|81.8|70.7|
> |LoReTrack-256|-|-|-|82.9| 81.4|-|70.3| 76.2| -|73.5|84.0| 70.4|
> |SeqTrack|56.8|66.8|45.6|83.3|88.3|82.2|69.9|79.7|76.3|74.7|84.7|71.8 |
> |AQATrack|-|-|-|83.8|88.6|**83.1**|**71.4**|**81.9**|**78.6**|73.8|83.2|72.1|
> |OSTrack|54.2|63.2|42.2| 83.1|87.8|82.0|69.1|78.7|75.2|71.0|80.4|68.2|
> |**DSATrack-ViT-B**|**60.2**|**69.1**|**50.2**|**84.1**|**88.6**|82.7|69.4|78.6|74.6|**75.0**|**85.6**|**73.7**|
> >***Q8**. While DSATrack outperforms SOTA models in speed, the paper lacks a detailed comparison of FLOPs and parameter*
> - We thank the reviewer for the suggestion. We conducted real-world experiments using newly bought drone and Jetson Xavier. The results demonstrate that our method achieves a balance between accuracy and speed, and surpasses previous methods in performance.
> |Method|DTB70-Pre.|DTB70-Suc.|UAVDT-Pre.|UAVDT-Suc.|VisDrone-Pre.|VisDrone-Suc.|UAV123-Pre.|UAV123-Suc.|Flops(G)|Params(M)|Xavier(FPS)|RTX3090(FPS)|
> |-|-|-|-|-|-|-|-|-|-|-|-|-|
> |HiT|75.1|59.2|62.3|47.1|74.8|58.7|82.5|63.3|4.35|42.14|36.6|159.0|
> |AVTrack-ViT|81.3|63.3|79.9|57.7|86.4|65.9|84.0|66.2|1.82|6.2|30.0|184.8|
> |LiteTrack|82.5|63.9|81.6|59.3|79.7|61.4|84.2|65.9|12.81|49.59|15.9|160.1|
> |**DSATrack-D7**|**88.1**|**68.4**|84.6|**62.1**|**88.6**|**68.0**|**87.3**|**66.6**|23.64|56.76|14.0|162.4|
> |DSATrack-D4|84.5|65.9|82.3|58.9|84.8|64.1|84.1|63.0|14.12|35.5|17.4|**214.1**|
> |DSATrack-ViT-T|86.6|67.4|84.3|61.6|84.8|64.1|85.5|65.5|2.99|8.21|27.6|114.3|
> |DSATrack-DeiT-T|84.5|65.7|**85.0**|62.0|82.0|60.8|84.4|64.5|2.99|8.21|27.4|114.4|
>
> >***Q9**. Failure Case Analysis.*
> - Based on your suggestion, we conducted a failure case analysis. We found that DSATrack performs poorly in out-of-view and full occlusion scenarios, where it struggles to re-detect and recover the target. This may be a common limitation faced by all window-based local trackers.
>
> - To address this issue, we plan to incorporate a two-stage re-detection mechanism in future work to recover lost targets.
>
> - As NeurIPS 2025 does not allow anonymous links or supplementary figures during the rebuttal phase, we will include these analyses and visualizations in the camera-ready. **Thank you for your suggestion—it is very helpful in improving the quality of our manuscript**.
>
> [1].Gao S, Zhou C, Zhang J. Generalized relation modeling for transformer tracking[C]//Proceedings of the IEEE/CVF conference on computer vision and pattern recognition. 2023: 18686-18695.
>
> [2]. Zhou X, Guo P, Hong L, et al. Reading relevant feature from global representation memory for visual object tracking[J]. Advances in Neural Information Processing Systems, 2023, 36: 10814-10827.

---

> > ### Comment · Reviewer_cKGP · 2025-08-05
> >
> > I would like to thank the authors for addressing my concerns. After reading the rebuttal, I tend to raise the score.

---

> > > ### Author Response · Authors · 2025-08-06
> > >
> > > Thank you very much for your thoughtful follow-up and raise the score. We truly appreciate your recognition of our efforts in addressing the concerns. we would be grateful if you could reflect that in your final rating and justification, it is a great encouragement and recognition for us. We sincerely thank you for your time, supportive review, and constructive feedback.

---

### Official Review · Reviewer_gMhJ · 2025-07-04

**Clarity:** 2
**Significance:** 3
**Originality:** 2
**Rating:** 4
**Confidence:** 3

**Summary:**

This paper presents DSATrack, a UAV tracking framework featuring a Dynamic Semantic Relevance Generator to improve semantic correlation modeling. A Transformer pruning strategy is also proposed to balance speed and accuracy. Experiments demonstrate state-of-the-art performance across multiple benchmarks.

**Questions:**

- How generalizable is DSATrack across different UAV tracking scenarios?

- Can the semantic-aware correlation maps be visualized to better demonstrate the interpretability of the method?
- More issues are shown in the weakness above.

**Ethical Concerns:**

["NO or VERY MINOR ethics concerns only"]

**Final Justification:**

Thanks for your detailed response. I'll raise the rating, as my concerns have been resolved.

**Limitations:**

yes

**Quality:**

3

**Strengths And Weaknesses:**

### Strengths
- The paper identifies an important gap in current UAV tracking methods—the lack of semantic awareness in correlation modeling, which is well-justified.

- By introducing a pruning-based variant generation, the method can flexibly adapt to different computational constraints (e.g., real-time vs. offline UAV applications).

- Extensive experimental results and ablation studies are conducted, proving the effectiveness of each proposed module.

### Weaknesses

- The details of the Dynamic Semantic Relevance Generator are insufficiently elaborated—how exactly it computes relevance, and how it integrates with the Transformer pipeline should be clarified.

- While pruning is introduced for speed-up, the comparison with existing efficient tracking baselines (e.g., lightweight Transformers or mobile-friendly trackers) is not reported.

- It's unclear how robust DSATrack is to extreme low-resolution or partial occlusion cases, which are critical for UAV applications.

---

> ### Author Rebuttal · Authors · 2025-07-29
>
> Thank you for your valuable suggestions. **During the rebuttal period, we have made significant efforts to address your concerns.**. Additionally, **we have provided detailed responses to your other questions. We sincerely hope that you will reconsider our paper**.
>
> >***Q1**. While pruning is introduced for speed-up, the comparison with existing efficient tracking baselines (e.g., lightweight Transformers or mobile-friendly trackers) is not reported.*
> - **Thank you for your valuable suggestion**. **We would like to clarify that all the tracking methods presented in our work are lightweight and efficient trackers**, as our target application is UAV tracking, which is typically constrained by limited onboard computational resources. Therefore, we focused exclusively on reporting lightweight tracking methods in our comparisons.
> - **To facilitate understanding and to avoid potential confusion, we have further categorized the trackers listed in Table 3** based on their type, architecture, and application field, as shown below:
> |**Method**|Type|Architecture|Field|
> |-|-|-|-|
> |**SiamAPN**|lightweight/Efficient Tracker| CNN-based Method|UAV Tracking|
> |**SiamAPN++**|lightweight/Efficient Tracker|CNN-based Method|UAV Tracking|
> |**HiFT**|lightweight/Efficient Tracker|CNN-based Method|UAVTracking|
> |**P-SiamFC++**|lightweight/Efficient Tracker|CNN-based Method|UAVTracking|
> |**TCTrack**|lightweight/Efficient Tracker|CNN-based Method|UAVTracking|
> |**UDAT**|lightweight/Efficient Tracker|CNN-based Method|UAVTracking|
> |**ABDNet**|lightweight/Efficient Tracker|CNN-based Method|UAVTracking|
> |**DRCI**|lightweight/Efficient Tracker|CNN-based Method|UAVTracking|
> |**HiT**|lightweight/Efficient Tracker|CNN-based Method|General Tracking|
> |**DDCTrack**|lightweight/Efficient Tracker|Transformer-based Method|UAV Tracking|
> |**SMAT**|lightweight/Efficient Tracker|Transformer-based Method|General Tracking|
> |**LiteTrack**|lightweight/Efficient Tracker|Transformer-based Method|General Tracking|
> |**AVTrack-ViT**|lightweight/Efficient Tracker|Transformer-based Method|UAV Tracking|
> |**AVTrack-DeiT**|lightweight/Efficient Tracker|Transformer-based Method|UAV Tracking|
>
> - For your reference, the performance comparison can be found in **Table 3**, and the speed comparison is presented in **Figure 5**.
> - **Once again, thank you for your valuable feedback. If you consider this table appropriate, I will incorporate it into the revised version of the manuscript in accordance with your suggestion**.
>
> >***Q2**. It's unclear how robust DSATrack is to extreme low-resolution or partial occlusion cases, which are critical for UAV applications.*
> - **Thank you for your valuable comment. We appreciate your attention** to evaluation under challenging scenarios. **To clarify, we have included the performance** of our model under low-resolution settings in **Table 6 of Appendix A.1**, which covers cases such as camera motion, fast motion, and low resolution.
>
> - Additionally, for your reference, **we have provided a comparison of precision with other state-of-the-art trackers under low-resolution and partial occlusion conditions in Figure 3**.
>
> - The results presented in Table 6 and Figure 3 demonstrate that our method **exhibits strong robustness under low-resolution and partial occlusion scenarios**. This can be attributed to our semantic-aware strategy, which enables more accurate feature alignment and consequently enhances overall model performance.
>
> >***Q3**. How generalizable is DSATrack across different UAV tracking scenarios?*
>  - **Thank you** for your valuable comment. **We would like to clarify that we have provided the precision performance comparison** across different UAV tracking scenarios in **Figure 3 and Figure 7 of Appendix A.1**, covering challenging scenarios such as **Low Resolution, Partial Occlusion, Full Occlusion, Camera Motion, Fast Motion, Background Clutters, Aspect Ratio Change,  Illumination Variation, Out-of-View, Scale Variation, Similar Object and Viewpoint Change**.
>
> - **Thank you for your comment**. To further clarify the results, **we create a table accodirng to Figure3 and Figure 7 of Appendix A.1** to facilitate your understanding and to avoid potential confusion. **If you find the table more appropriate, we will replace the Figure 3 and Figure 7 of Appendix A.1 with the table in the revised manuscript to present the results across different scenarios more clearly**.
>  |**Method**|Low Resolution|Partial Occlusion|Fast Motion|Camera Motion|Full Occlusion|Background Clutter |Aspect Ratio Change | Out-of-View|
> |-|-|-|-|-|-|-|-|-|
> |**ECO-HC**|54.7|79.6|67.9|80.0|75.1|64.9|88.3|87.0|
> |**SiamAPN++**|58.8|69.8|75.9|70.8|64.4|51.9|81.2|81.4|
> |**BACF**|59.8|71.8|57.3|77.3|67.1|60.2|73.7|81.2|
> |**KCF**|64.2|62.1|56.9|70.0|58.1|52.8|62.9|58.2|
> |**TCTrack**|64.4|76.0|73.8|78.5|71.7|62.4|86.0|87.9||
> |**AVTrack**|73.8|79.2|82.7|86.5|80.3|73.0|92.1|91.0|
> |**DSATrack-ViT-B**|74.5|82.8|74.5|85.8|78.1|74.6|**91.7**|94.8|96.4|
> |**DSATrack-ViT-T**|77.1|78.4|76.5|86.3|77.9|76.0|**95.9**|94.1|
> |**DSATrack-D8**|86.8|**86.3**|**88.4**|**89.7**|**82.3**|**80.4**|93.5|96.4|
> |**DSATrack-D7**|**87.2**|82.2|82.0|87.0|76.2|78.0|94.7|96.9|
> |**DSATrack-D6**|76.2|82.1|85.9|87.7|81.3|76.5|85.9|86.2|
> |**DSATrack-D4**|80.6|77.4|86.0|87.6|79.0|74.0|88.4|89.0|
>
> >***Q4**. Can the semantic-aware correlation maps be visualized to better demonstrate the interpretability of the method?*
> - Thank you for your valuable comment.  The visualization of semantic-aware correlation maps has been presented in Figure 4. In each row of Figure 4, (a) shows the input frame, while (b) and (c) depict the correlation maps before and after applying the semantic correlation module, respectively. Brighter areas indicate stronger similarity. The enhanced focus and localization in (c) demonstrate the effectiveness of our semantic-aware module under challenging conditions such as camera motion, fast motion, and low resolution.
>
> >***Q5**. The details of the Dynamic Semantic Relevance Generator are insufficiently elaborated—how exactly it computes relevance, and how it integrates with the Transformer pipeline should be clarified.*
>
> - Thank you for your constructive comment. We now provide a detailed clarification of how the **Dynamic Semantic Relevance Generator (DSRG)** computes relevance and integrates with the Transformer pipeline. Please refer to **Section 3.2** and **Figure 2** of the paper for visual references, and to **Equations (1)–(13)** for full definitions.
>
> ---
>
> - ### (1) Initial Correlation Map Computation
> - We begin by computing a high-dimensional correlation tensor between search queries and template keys:
> $
> C_{zx} = \frac{Q_x K_z^\top}{\sqrt{d_k}} \in \mathbb{R}^{h_x \times w_x \times h_z \times w_z \times l}
> $
> This represents dense similarity scores between all pairs of tokens. (See Equation (1), Line 120)
>
> ---
>
> - ### (2) Dynamic Semantic Relevance Estimation
>
> - We reshape $$
> C_{zx} \in \mathbb{R}^{h_x \times w_x \times h_z \times w_z \times l}
> \ \rightarrow \
> C_{zx} \in \mathbb{R}^{N_x \times N_z \times l}, \quad
> N_x = h_x \cdot w_x, \quad N_z = h_z \cdot w_z \tag{1}
> $$.
> Then we:
> Construct a semantic graph \( G = (V, E) \), where each node corresponds to a template token.
> Apply average pooling and concatenate features into a vector matrix $$\vec{v} \in \mathbb{R}^{N_z \times l}$$:
> $$
> \vec{v}\_i = \frac{1}{N\_x} \sum\_{j=1}^{N\_x} C\_{zx}[j, i, :]\tag{2}
> $$
> Compute pairwise semantic similarity matrix:
> $
> A = \vec{v} \cdot \vec{v}^\top \in \mathbb{R}^{N_z \times N_z \times l} \tag{3}
> $
> Estimate relevance scores using an MLP and LogSoftmax:
> $$
> \pi_{ij} = \text{LogSoftmax}(\text{MLP}(A_{ij})) \in \mathbb{R}^{2} \tag{4}
> $$
> Use Gumbel-Softmax to dynamically generate a binary semantic relevance mask:
> $$
> E = \text{GumbelSoftmax}(\pi) \in \{0,1\}^{N_z \times N_z} \tag{5}
> $$
> (See Equations (2)–(5), Lines 131–139; and Figure 2, bottom-left panel.)
>
> ---
>
> - ### (3) Refined Semantic-Aware Correlation Map
> - We propagate correlation features across semantically relevant nodes using graph convolution with normalization:
> $C^{\prime}_{zx} = \Lambda^{-1/2} \hat{E} \Lambda^{-1/2}C^\top\_{zx} W_v \in \mathbb{R}^{N_x \times N_z \times l} \tag{6}
> $
> Where $\hat{E} = E + I $, $I$ is the identity matrix and $ \Lambda $ is the degree matrix.
> (See Equation (7), Line 141)
>
> ---
>
> - ### (4) Integration into Transformer Pipeline
> - The refined semantic-aware correlation map $( C'_{zx} )$ is used in two ways:
> - **Token Filtering** (Eq. (8) Line 145): Discards less informative template tokens based on MLP + Gumbel-Softmax decisions.
> **Hybrid Attention Module** (Eq. (9)–(11) Line 146-148): Combines Self-attention on both search and template tokens, and Cross-attention from search to template using both $C_{zx}$  and $ C'_{zx}$ :
> $$
> \text{CrossAttention}(Q\_x, K\_z, V\_z) = \text{Softmax}(C\_{zx} + C'\_{zx}) V_z \tag{7}
> $$
> The final updated tokens are passed through a feedforward network:
> $$
> T''\_{xz} = T'\_{xz} + \text{FFN}(T'\_{xz}), \quad T'\_{xz}=\text{HybridAttention}(\cdot) + T\_{xz} \tag{8}
> $$
> (See Equations (8)–(13), Lines 143–152; and Figure 2, bottom-right block.)
>
> ---
>
> - ### (5) Summary
> - The **DSRG** dynamically refines semantic relevance via learnable attention.
> It is inserted at selected Transformer layers (layers 4, 7, 10).
> The resulting semantic-aware correlation map helps eliminate background noise and improves attention focus.
> Its benefit is validated by:
>   - Ablation study in **Table 4** and **Table 6**,
>   - Visual evidence in **Figure 4**.
>
> We hope this clarified explanation helps the reviewer understand the motivation, computation, and integration of the proposed DSRG module. Thank you again for your valuable feedback.

---

### Note · Authors · 2025-08-14

We sincerely appreciate the reviewers for acknowledging our clarifications during the rebuttal period and for recognizing the value of our work. We are also grateful to the PC, SAC, and AC for their hard work and dedication. Thank you.

---

### Decision · Program_Chairs · 2025-09-17

**Decision:**

Accept (poster)

**Comment:**

Paper Summary:
This paper introduces DSATrack, a UAV tracking framework that uses a novel Dynamic Semantic Relevance Generator to improve semantic-aware correlation modeling. The authors also propose a Transformer pruning strategy to create model variants that balance state-of-the-art performance with real-time speed requirements.

Main Strengths and Weaknesses:
The primary strength is its state-of-the-art performance across multiple UAV tracking benchmarks, stemming from its well-designed semantic-aware correlation modeling and comprehensive experiments. Its main initial weakness, cited by multiple reviewers, was a lack of clarity in explaining the core mechanisms and an insufficient comparison to a broader range of contemporary tracking methods.

Rebuttal Analysis:
The rebuttal effectively addresses the majority of concerns raised by all four reviewers. After the rebuttal, three of the four reviewers explicitly raised their ratings or scores, citing that their questions about methodological clarity, experimental comparisons, and technical details had been satisfactorily resolved. The fourth reviewer, who had already given a positive score, confirmed that the rebuttal addressed most of their concerns.

Final Justification:
There is a clear and positive consensus among all reviewers. While the initial reviews highlighted legitimate concerns regarding clarity and the completeness of the evaluation, the responses of the authors successfully mitigated these issues. Considering the interesting idea and the state-of-the-art results on several relevant benchmarks, the AC agrees with the reviewers and recommends accepting the paper. Additionally, it is strongly suggested that the authors incorporate the content discussed in the feedback to make the paper more comprehensive.